# RoomEditor: High-Fidelity Furniture Synthesis with Parameter-Sharing U-Net

**Zhenyi Lin**
Tianjin University
linzhenyi@tju.edu.cn

**Xiaofan Ming**
Tianjin University
xiaofanming@tju.edu.cn

**Qilong Wang**
Tianjin University
qlwang@tju.edu.cn

**Dongwei Ren**[*]
Tianjin University
rendw@tju.edu.cn

**Wangmeng Zuo**
Harbin Institute of Technology
wmzuo@hit.edu.cn

**Qinghua Hu**
Tianjin University
huqinghua@tju.edu.cn

Indoor scene

General scene

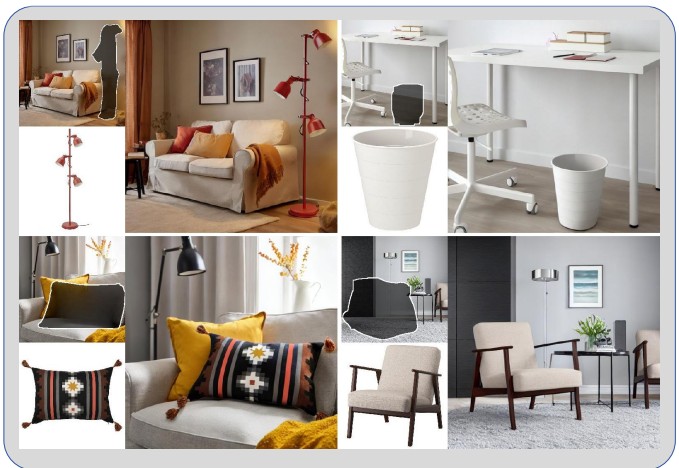
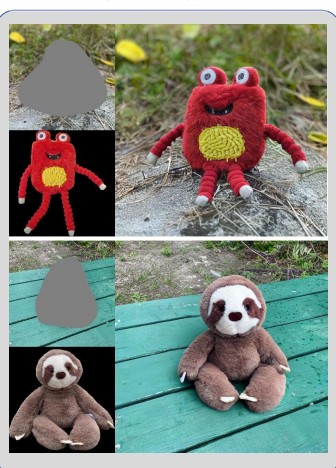

Figure 1: Furniture synthesis with RoomEditor integrates reference objects into environments with geometric coherence and visual fidelity. Moreover, RoomEditor exhibits remarkable generalization capabilities across a wide range of unseen scenes and objects without task-specific fine-tuning

## Abstract

Virtual furniture synthesis, a critical task in image composition, aims to seamlessly integrate reference objects into indoor scenes while preserving geometric coherence and visual realism. Despite its significant potential in home design applications, this field remains underexplored due to two major challenges: the absence of publicly available and ready-to-use benchmarks hinders reproducible research, and existing image composition methods fail to meet the stringent fidelity requirements for realistic furniture placement. To address these issues, we introduce RoomBench, a ready-to-use benchmark dataset for virtual furniture synthesis, comprising 7,298 training pairs and 895 testing samples across 27 furniture categories. Then, we propose RoomEditor, a simple yet effective image composition method that employs a parameter-sharing dual U-Net architecture, ensuring better feature consistency by sharing weights between dual branches. Technical analysis reveals that conventional dual-branch architectures generally suffer from inconsistent intermediate features due to independent processing of reference and background images. In contrast, RoomEditor enforces unified feature learning through shared parame-

---

[*]Corresponding author

39th Conference on Neural Information Processing Systems (NeurIPS 2025).

ters, thereby facilitating model optimization for robust geometric alignment and maintaining visual consistency. Experiments show our RoomEditor is superior to state-of-the-arts, while generalizing directly to diverse objects synthesis in unseen scenes without task-specific fine-tuning. Our dataset and code are available at https://github.com/stonecutter-21/roomeditor.

# 1 Introduction

Recent advances in augmented reality and computer vision have revolutionized virtual product visualization for e-commerce applications, particularly in indoor scene synthesis. Within the rapidly expanding home design market (expected to reach around 250 billion dollars in ten years [15]), intelligent image synthesis systems have emerged as critical tools for consumer decision-making. These systems enable in-situ furniture visualization by digitally integrating selected items into user-provided room images as shown in Figure 1, offering significant potential to transform design practices and retail experiences.

Despite progress in generic image composition, indoor furniture synthesis remains underdeveloped due to two key limitations. The first lies in the limited availability of ready-to-use datasets with realistic indoor environments, thereby hindering reproducible research. While Amazon's dataset [44] demonstrates the value of domain-specific data, its restricted access stifles community efforts. InteriorNet [30], 3D-FUTURE [14], and 3D-FRONT [13] expose detailed 3D models, but constructing reasonable reference–background pairs demands meticulous viewpoint selection and scene setup, revealing challenges in data acquisition. Consequently, the field remains hampered by inadequate benchmarks and persistent data scarcity. To address this issue, we introduce the RoomBench dataset, a ready-to-use, publicly available benchmark tailored for virtual furniture synthesis. Our RoomBench consists of 7,298 training pairs and 895 testing samples across 27 furniture categories, meticulously curated from diverse online sources. Each sample includes high-quality annotations, enabling comprehensive evaluation of geometric coherence and visual realism. This benchmark provides a foundation for future research in indoor furniture synthesis by facilitating both training and evaluation.

The second critical challenge arises from technical limitations in current image composition methods [53, 6, 7], which often fall short of the high visual fidelity required in practical home design applications. Minor geometric misalignments or texture inconsistencies can break user immersion and erode trust. Image encoder-based methods such as PBE [53] and Anydoor [6] use CLIP [38] or DINO [3] to extract reference features and inject them into the U-Net. However, they tend to overlook fine-grained structural details essential for complex furniture integration. Even advanced dual U-Net architectures like MimicBrush [7] struggle to ensure visual consistency and harmony between the inserted object and the surrounding background, often resulting in unnatural compositions with obvious copy-paste artifacts. These limitations result in inconsistent appearances and poses of furniture, thereby undermining practical utility. To address these limitations, we propose RoomEditor, a parameter-sharing dual U-Net that enforces feature consistency via shared weights. We perform an in-depth analysis of existing dual U-Net pipelines and identify feature misalignment between the extraction and inpainting stages, which degrades synthesis quality. By unifying feature extraction and inpainting within the same network, RoomEditor naturally aligns the two stages, enabling complex spatial transformations and seamless object integration. Experiments demonstrate that our method achieves superior objective and subjective results on RoomBench compared to state-of-the-art methods.

Our RoomEditor not only excels in home design but also generalizes effectively to a broader range of scenes and objects. As illustrated in Figure 1, our RoomEditor can be directly applied to general scenes and objects without any task-specific fine-tuning. The superior adaptability of RoomEditor can be attributed to its unified feature learning in parameter-shared U-Net. The contributions of this work are summarized as follows:

- We constructed a ready-to-use open-source benchmark RoomBench specifically for home design, consisting of 7,298 furniture-background training pairs and 895 testing samples across 27 categories with annotations.

- We propose a simple yet effective RoomEditor for high-fidelity furniture synthesis, whose core is a parameter-sharing U-Net. In addition, we provide an in-depth analysis of existing

dual-branch approaches from the perspective of feature consistency, showing the potential advantage of RoomEditor for high-fidelity and visual consistency.

- Extensive experiments validate that RoomEditor achieves superior performance in both quantitative and qualitative evaluations as well as in human perception studies for home design, while demonstrating remarkable generalization across diverse scenes and objects.

## 2 Related Work

### 2.1 Furniture Synthesis for Home Design

The most relevant research field to furniture synthesis is virtual try-on [18, 8, 28, 26, 50, 60], which has primarily focused on fashion applications, emphasizing fabric textures, body pose estimation, and occlusion handling. While several virtual try-on methods [26, 50, 60] employ dual U-Net architectures and must model garment–body interactions, their objectives and challenges differ from those in furniture synthesis. In particular, furniture synthesis requires handling diverse and rigid object shapes, whereas clothing in virtual try-on typically exhibits more consistent forms, leading to different integration challenges. Meanwhile, the area of indoor scene synthesis specifically for furniture placement remains relatively underexplored. A few studies [4, 49] have explored interior design through text-to-image generation but provide limited control over individual furniture placement. Diffuse-to-Choose [44] addresses furniture synthesis by incorporating reference features via a U-Net encoder augmented with FiLM [37] layers on a large-scale indoor dataset. However, its training and testing sets are proprietary, limiting reproducibility and further research. To bridge these gaps, this paper introduces RoomBench, a ready-to-use and open-source benchmark for furniture synthesis, along with a method RoomEditor that ensures high-fidelity object preservation and context-aware placement.

### 2.2 Image Composition

Early image composition methods [23, 9, 45, 46, 47, 11, 10] primarily focused on the task of image harmonization, ensuring seamless integration of the foreground with the background. These methods typically relied on manually designed pipelines. In contrast, text-to-image diffusion models [40, 39, 17, 42] have enabled the automatic integration of specific objects into diverse contexts while preserving their identity and attributes. By extracting pseudo-words, methods such as Textual Inversion [16] and DreamBooth [41], along with others [2, 19, 24, 25, 31, 33, 12, 29, 27], rely on fine-tuning with textual prompts but often produce unstable backgrounds. Semantic image composition methods, such as DreamPaint [43], PBE [53], CustomNet [54], and ControlCom [55], which focus on object insertion into predefined scenes, often struggle to preserve fine-grained details in complex scenes when using CLIP [38] image encoder. Recent advancements, including AnyDoor [6], have addressed some of these challenges by leveraging ControlNet [57] and DINO [35] to improve texture fidelity and reference-object consistency. In contrast, MimicBrush [7] employs a dual U-Net architecture [51, 56, 59, 5, 22, 52], which has proven effective in capturing multi-scale reference features, thereby enabling flexible image editing and generating high-fidelity syntheses. Despite these improvements, achieving both high fidelity and seamless integration in home design remains a significant challenge, particularly when the training dataset is insufficient.

## 3 Benchmark for Furniture Synthesis

In this section, we construct an open-source RoomBench for furniture synthesis, whose procedures for data collection, filtering, and annotation are detailed as follows.

### 3.1 Data Collection and Filtering

To ensure the diversity in furniture types and scene contexts, we systematically collected about 4,500 furniture items, each of which has several product images and contextual scene backgrounds. Our curation strategy included two key measures. Category Coverage: Hierarchical classification into 27 fine-grained categories, including bedroom essentials (e.g., Bed, Pillow), living room furnishings (e.g., Sofa, Coffee Table), and bathroom fixtures (e.g., Shower, Bathtub). Image Quality Assurance: We require a minimum resolution of $512 \times 512$ for both furniture images and background images.

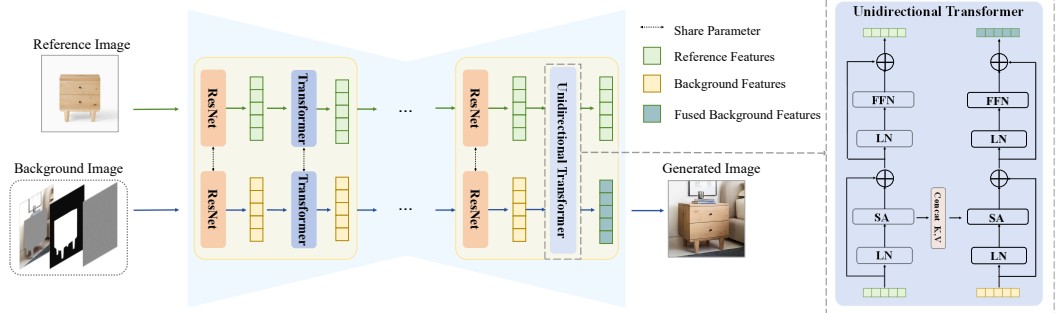

Figure 2: **The architecture of our RoomEditor.** Our method shares parameters between the two U - Nets for unified feature space learning. As shown, reference features propagate independently, while background features interact with reference features through a self - attention module at each layer, ensuring effective feature alignment.

Subsequently, the collected data underwent rigorous filtering. Initially, we manually classified images as either product images or background images. We then leveraged GPT-4o [1] to assist in data filtering. Images were excluded if the furniture was not fully visible in the background image or if there was an inconsistency between the furniture object or color in the reference and background images (i.e., they did not depict the same object), resulting in 5,948 reference furniture images and 4,989 background images. Following this, the dataset was split into training and testing sets. The training set comprises 5,288 reference furniture images and 4,094 background images, with each reference image paired with one or two background images, resulting in a total of 7,298 furniture–background pairs. The testing set contains 660 reference furniture images and 895 background images, resulting in 895 testing samples. For more detailed data-processing procedures and complete dataset information, please refer to Appendix B.

### 3.2 Data Annotation

For each image pair in RoomBench, background image and reference image are annotated with corresponding furniture masks. In the training set, these masks are hand-labeled precisely to ensure high-quality annotations. This labeling process allows effective data augmentation, improving the model's generalization capabilities. For the testing set, we included more loosely annotated masks to reflect the variability and potential inaccuracies in user-generated labels. This annotation strategy allows us to evaluate the model's robustness in real-world scenarios, where annotations are often imperfect or inconsistent. With specific images and annotations, RoomBench is immediately ready-to-use, eliminating the need for additional efforts in data acquisition that are typically required by InteriorNet [30], 3D-FUTURE [14], and 3D-FRONT [13].

## 4 RoomEditor for Furniture Synthesis

In this section, we first introduce the RoomEditor architecture, whose core is a shared U-Net model for background and reference images with unidirectional information flow. Then, we compare our RoomEditor with existing methods from the perspective of feature consistency, showing the underlying mechanism behind our RoomEditor.

### 4.1 RoomEditor Architecture

Given a masked background image $I_{\text{bg}}^M$ and a masked reference image $I_{\text{ref}}^M$, our goal is to seamlessly integrate the object from $I_{\text{ref}}^M$ into the masked region of $I_{\text{bg}}^M$. Different from existing dual U-Net methods (e.g., [7]), our approach utilizes a shared U-Net to synchronously extract reference image features and perform inpainting. Specifically, as shown in Figure 2, we employ an inpainting U-Net $\epsilon_\theta(\cdot)$ as basic model by following previous work [7]. First, we extract the features for $I_{\text{ref}}^M$ from the mid- and upper-attention layers of the U-Net. For $l$-th self-attention layer, the extracted feature of

$I_{\text{ref}}^M$, denoted as $f_l(I_{\text{ref}}^M)$, is computed as

$$f_l(I_{\text{ref}}^M) = \text{softmax}\left(\frac{Q_{\text{ref}} \cdot K_{\text{ref}}^{\top}}{\sqrt{d_k}}\right) \cdot V_{\text{ref}}, \tag{1}$$

where $Q_{\text{ref}} = f_{l-1}(I_{\text{ref}}^M)W_Q$, $K_{\text{ref}} = f_{l-1}(I_{\text{ref}}^M)W_K$, $V_{\text{ref}} = f_{l-1}(I_{\text{ref}}^M)W_V$, and $W_{\{Q,K,V\}}$ are the projection matrices in attention layer. $f_{l-1}(I_{\text{ref}}^M)$ indicates the output of the $(l\text{-}1)$-th layer. It is worth note that our RoomEditor actually takes the image, mask and noise as inputs, while we omit the mask and noise in our formulation for simplicity.

Then, RoomEditor performs a unidirectional interaction to inject features of reference image $f_{l-1}(I_{\text{ref}}^M)$ into those of masked background image $I_{\text{bg}}^M$ for inpainting. Particularly, the features of $I_{\text{bg}}^M$ in $l$-th layer is computed by a mixture attention with inputs of $f_{l-1}(I_{\text{bg}}^M)$ and $f_{l-1}(I_{\text{ref}}^M)$, i.e.,

$$f_l(I_{\text{bg}}^M) = \text{softmax}\left(\frac{Q_{\text{bg}} \cdot K_{[\text{bg},\text{ref}]}^{\top}}{\sqrt{d_k}}\right) \cdot V_{[\text{bg},\text{ref}]}, \tag{2}$$

where $Q_{\text{bg}} = f_{l-1}(I_{\text{bg}}^M)W_Q$, $K_{[\text{bg},\text{ref}]} = \text{cat}(f_{l-1}(I_{\text{bg}}^M), f_{l-1}(I_{\text{ref}}^M))W_K$, $V_{[\text{bg},\text{ref}]} = \text{cat}(f_{l-1}(I_{\text{bg}}^M), f_{l-1}(I_{\text{ref}}^M))W_V$ and cat indicates the concatenation operation. Besides, existing dual U-Net methods [7, 50] generally integrate reference features by using extra image encoders (e.g., CLIP [38]). Due to similar functions are achieved by a mixture attention in Eqn. (2), our RoomEditor can omit extra image encoders to make our architecture more succinctly.

For optimizing our RoomEditor, we first construct masked background as $I_{\text{bg}}^M = M_{\text{bg}} \odot I_{gt}$ by giving ground-truth image $I_{\text{gt}}$ and a mask $M_{\text{bg}}$, and then sample random noise $\epsilon$ and a timestep $t$ to generate the noisy image $I_{gt}^t$. Finally, we minimize the following loss function, i.e.,

$$\mathcal{L} = \mathbb{E}_{t,I_{\text{gt}},\epsilon}\left\|\epsilon_\theta(I_{\text{gt}}^t, I_{\text{bg}}^M, c, t) - \epsilon\right\|_2^2, \tag{3}$$

where $c$ denotes the features extracted from $I_{\text{ref}}^M$.

## 4.2 Discussion on Merit of RoomEditor

In this subsection, we compare with previous works [53, 6, 7] from the perspective of feature consistency, and show the potential advantage of our RoomEditor architecture in the context of high-fidelity furniture synthesis task.

### 4.2.1 Task Description

Since high-fidelity furniture synthesis aims to prominently place reference furniture $I_{\text{ref}}$ in the background image $I_{\text{bg}}$, it can be formulated as an ideal copy-paste problem. Specifically, the target image $I_{\text{gt}}$ can be represented as a composition of a masked background image $M_{\text{bg}} \odot I_{\text{bg}}$ and a masked reference image $M_{\text{ref}} \odot I_{\text{ref}}$ with pose transformation $\mathcal{R}$, i.e., $I_{\text{gt}} = M_{\text{bg}} \odot I_{\text{bg}} + \mathcal{R}(M_{\text{ref}} \odot I_{\text{ref}})$, where $\odot$ indicates element-wise multiplication. Therefore, the key issue is learning the precise transformation $\mathcal{R}$, given a pair of background and reference images $\{I_{\text{bg}}, I_{\text{ref}}\}$. As $\mathcal{R}$ depends on the spatial and visual relationship between $I_{\text{bg}}$ and $I_{\text{ref}}$, including factors such as mask geometry and background illumination, the final objective can be formulated as

$$I_{\text{gt}} = M_{\text{bg}} \odot I_{\text{bg}} + \mathcal{R}(M_{\text{ref}} \odot I_{\text{ref}} \mid M_{\text{bg}} \odot I_{\text{bg}}). \tag{4}$$

To learn transformation $\mathcal{R}$, diffusion model $\epsilon_\theta(\cdot)$ is generally employed to achieve this implicitly by training on abundant pairs of background and reference images $\{I_{\text{bg}}, I_{\text{ref}}\}$ with masks $\{M_{\text{bg}}, M_{\text{ref}}\}$.

Based on the empirical observation that inpainting models implicitly preserve unmasked regions during training, we treat predictions in these areas as ground truth. Accordingly, we consider two complementary cases: (1) masking the object region in $I_{\text{gt}}$ to obtain $I_{\text{bg}}^{\text{M}}$, where the background prediction remains unchanged; and (2) masking the background region to retain only the object (denoted as $\mathcal{R}(I_{\text{ref}}^{\text{M}} \mid I_{\text{bg}}^{\text{M}})$), where the object prediction remains unchanged. Combining these two masked predictions yields an approximation of the target image:

$$\begin{aligned}
\epsilon_\theta(I_{\text{gt}}) &= \epsilon_\theta\left(M_{\text{bg}} \odot I_{\text{bg}} + \overline{M_{\text{bg}}} \odot \mathcal{R}\left(I_{\text{ref}}^{\text{M}} \mid I_{\text{bg}}^{\text{M}}\right)\right) \\
&\approx M_{\text{bg}} \odot \epsilon_\theta\left(I_{\text{bg}}^{\text{M}}\right) + \overline{M_{\text{bg}}} \odot \epsilon_\theta\left(\mathcal{R}\left(I_{\text{ref}}^{\text{M}} \mid I_{\text{bg}}^{\text{M}}\right)\right),
\end{aligned} \tag{5}$$

where $I_{\text{bg}}^{\text{M}}$ and $I_{\text{ref}}^{\text{M}}$ represent $M_{\text{bg}} \odot I_{\text{bg}}$ and $M_{\text{ref}} \odot I_{\text{ref}}$, respectively, and $\overline{M_{\text{bg}}} = \mathbf{1} - M_{\text{bg}}$ denotes the complement of the mask $M_{\text{bg}}$.

Similarly, since the inpainting model's output in non-masked regions remains essentially unchanged regardless of input conditions, its impact at the feature level is marginal. Thus, for the $l$-th layer, we obtain

$$
\begin{aligned}
f_l(I_{\text{gt}}) &= f_l \left( M_{\text{bg}} \odot I_{\text{bg}} + \overline{M_{\text{bg}}} \odot \mathcal{R} \left( I_{\text{ref}}^{\text{M}} \mid I_{\text{bg}}^{\text{M}} \right) \right) \\
&\approx M_{\text{bg}} \odot f_l \left( I_{\text{bg}}^{\text{M}} \right) + \overline{M_{\text{bg}}} \odot f_l \left( \mathcal{R} \left( I_{\text{ref}}^{\text{M}} \mid I_{\text{bg}}^{\text{M}} \right) \right),
\end{aligned}
\tag{6}
$$

Furthermore, the diffusion model implicitly learns $\mathcal{R}$ by progressively transforming and fusing features at each layer, rather than by directly transforming the original image. Thus, Eqn. (6) can be rewritten as follows in practice:

$$
f_l(I_{\text{gt}}) \approx M_{\text{bg}} \odot f_l \left( I_{\text{bg}}^{\text{M}} \right) + \overline{M_{\text{bg}}} \odot \mathcal{R}_l \left( f_l \left( I_{\text{ref}}^{\text{M}} \right) \mid f_l \left( I_{\text{bg}}^{\text{M}} \right) \right).
\tag{7}
$$

where $\mathcal{R}_l$ denotes the transformation in feature space. Given that the first term is solely related to the background, the differences among various methods [53, 6, 7] are primarily concentrated on the second term. According to Eqn. (6) and Eqn. (5), better feature consistency between ground-truth $f_l(I_{\text{gt}})$ and practice one (i.e., the right part of Eqn. (6)) will help to optimize the inpainting model in Eqn. (5). The experimental verification of Eqn. (5) and Eqn. (6) is presented in subsection C.1.

### 4.2.2 Comparison with Previous Works

From the perspective of feature consistency, we compare our RoomEditor with several previous works [53, 6, 7]. Here we consider the $l$-th interaction layer and show how different methods approximate $f_l(I_{\text{gt}})$. Encoder-based methods [53, 6] take advantage of a frozen image encoder $E(\cdot)$ and a trainable linear projector $L(\cdot)$ to extract reference features, formulated as $E_L(I_{\text{bg}}^{\text{M}}) = L(E(I_{\text{bg}}^{\text{M}}))$. Then, cross-attention is applied to inject the reference features $L(E(I_{\text{ref}}^{\text{M}})) \in \mathbb{R}^{T' \times d}$ into the background ones:

$$
\widehat{f_l(I_{\text{gt}})} = \text{Cross-Attn}(f_{l-1}(I_{\text{bg}}^{\text{M}}), E_L(I_{\text{bg}}^{\text{M}})),
\tag{8}
$$

where $\text{Cross-Attn}(\cdot, \cdot)$ denotes cross-attention.

Dual U-Net methods (e.g., [7]) employ a frozen U-Net $g_{l-1}(\cdot)$ to extract attention features at $(l\text{-}1)$-th layer. These features are injected via mixture attention (which can be approximated as a blend of self- and cross-attention), expressed as

$$
\widehat{f_l(I_{\text{gt}})} \approx \underbrace{\text{Self-Attn}(f_{l-1}(I_{\text{bg}}^{M}))}_{f_l(I_{\text{bg}}^{M})} + \underbrace{\text{Cross-Attn}(f_{l-1}(I_{\text{bg}}^{M}), g_{l-1}(I_{\text{ref}}^{M}))}_{A_{[\text{bg,ref}]}(g_{l-1}(I_{\text{ref}}^{\text{M}})) \cdot W_V},
\tag{9}
$$

where $\text{Self-Attn}(\cdot, \cdot)$ denotes self-attention and $A_{[\text{bg,ref}]}$ denotes attention score matrix. Compared to Eqn. (7), these methods approximate $\mathcal{R}_l(f_l(I_{\text{ref}}^{\text{M}}) \mid f_l(I_{\text{bg}}^{\text{M}}))$ through the operation $A_{[\text{bg,ref}]}(g_{l-1}(I_{\text{ref}}^{\text{M}})) \cdot W_V$, while $M_{\text{bg}}$ and $\overline{M_{\text{bg}}}$ are implicitly learned through the attention mechanism.

In contrast, our RoomEditor uses a single U-Net $f_{l-1}(\cdot)$ to extract reference features which are integrated through mixture attention:

$$
\widehat{f_l(I_{\text{gt}})} \approx \underbrace{\text{Self-Attn}(f_{l-1}(I_{\text{bg}}^{\text{M}}))}_{f_l(I_{\text{bg}}^{\text{M}})} + \underbrace{\text{Cross-Attn}(f_{l-1}(I_{\text{bg}}^{\text{M}}), f_{l-1}(I_{\text{ref}}^{\text{M}}))}_{A_{[\text{bg,ref}]}(f_{l-1}(I_{\text{ref}}^{\text{M}})) \cdot W_V}.
\tag{10}
$$

Compared to Eqn. (7), our method approximates $\mathcal{R}_l(f_l(I_{\text{ref}}^{\text{M}}) \mid f_l(I_{\text{bg}}^{\text{M}}))$ using $A_{[\text{bg,ref}]}(f_{l-1}(I_{\text{ref}}^{\text{M}})) \cdot W_V$. Under this formulation, we assume $\mathcal{R}_l$ corresponds to a cross-attention operation, which applies a complex transformation to the reference features based on their similarity to the background features. It is worth note that the optimal $g_l(\cdot)$ may be not $f_l(\cdot)$, but $f_l(\cdot)$ is a natural choice for offering the promising performance. Empirical comparisons in terms of loss visualizations and image quality indicate that the shared U-Net outperforms both the frozen U-Net and the unshared one.

**Empirical Validation.** To further investigate the effect of feature consistency, we compare three different methods on our RoomBench, i.e., frozen reference U-Net as in Mimicbrush [7], trainable reference U-Net for reference images, and our RoomEditor with shared U-Net for background and reference images. Specifically, we train all models on our RoomBench under the same exactly setting,

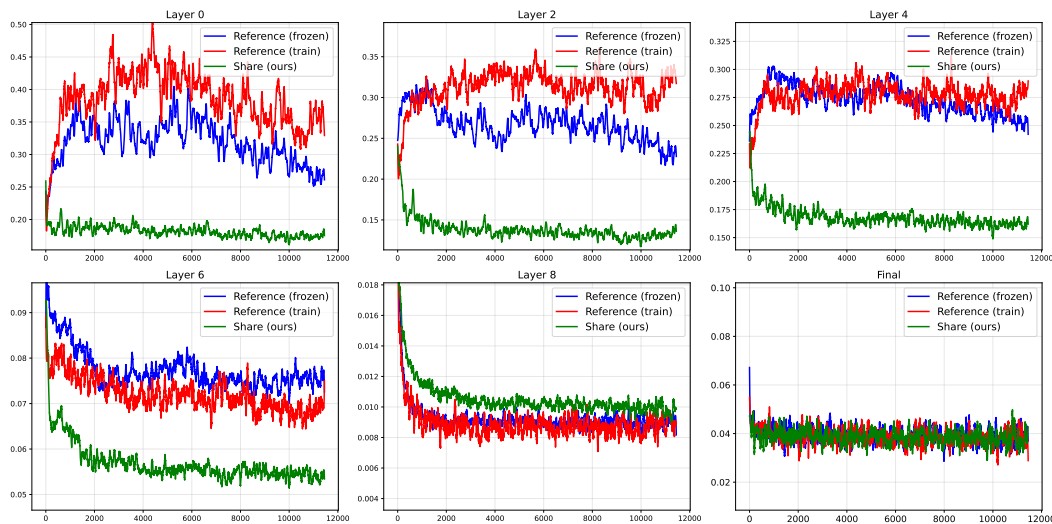

Figure 3: $\ell_2$ error values in different layers by three methods: our RoomEditor (green line), and two non-shared dual U-Net methods with frozen (blue line) and trainable (red line) reference U-Net.

and compute $\ell_2$ loss between features from two separate inputs $(\widehat{f_l(\boldsymbol{I}_{\text{gt}})})$ and those of the ground-truth images, which are outputted by the intermediate layers of inpainting U-Net $(f_l(\boldsymbol{I}_{\text{gt}}))$. Figure 3 shows the results of different methods along with various layers and training epochs, where we can observe (1) our RoomEditor with shared U-Net (green line) is consistently low in $\ell_2$ loss across all intermediate layers. Particularly, RoomEditor is significantly lower than other two methods at the early stages in terms of $\ell_2$ loss. (2) The frozen (blue line) and trainable (red line) reference U-Net only approximate the features of ground-truth images at the last stage under the strong constraint of training loss. (3) The frozen reference U-Net has lower $\ell_2$ loss than trainable one in the early stages, indicating unshared U-Net is more difficult to optimize, due to large parameter space and limited training data. The above comparisons show that our RoomEditor, using a shared U-Net, achieves better feature consistency, leading to better performance (refer to experiments in Table 4).

# 5 Experiments

## 5.1 Experiment Setup

**Implementation Details.** Our RoomEditor is initialized with the Stable Diffusion 1.5 inpainting model [40] and trained on the RoomBench dataset for 20k steps with a batch size of 32 across four NVIDIA A6000 GPUs. We use the AdamW optimizer [32] with a constant learning rate of $1 \times 10^{-5}$, and set the input resolution to $512 \times 512$. To enhance robustness against imperfect mask inputs and domain variations, we apply augmentation techniques to both masks and images. The augmented masks are shown in Figure 6.

**Objective Evaluation Metrics.** To compare with other methods, we use SSIM [48] and PSNR [21] for assessing reconstruction quality, FID [20] and LPIPS [58] for assessing perceptual realism, CLIP-score [38] and DINO-score [35] for assessing semantic consistency.

**Human Evaluation Metrics.** To complement objective metrics, we conducted a user study following [7], evaluating results in fidelity (retaining reference identity and details), harmony (seamless integration with the background), and quality (overall visual appeal and detail).

## 5.2 Evaluation on RoomBench

Our RoomEditor is compared with PBE [53], Anydoor [6] and Mimicbrush [7]. These competing methods are evaluated under three settings: the released models trained on their respective large datasets, the fine-tuned models (denoted with ⋆), whose released model weights are further fine-tuned

| Method | Train Data[‡] | RoomBench | FID↓ | SSIM↑ | PSNR↑ | LPIPS↓ | CLIP↑ | DINO↑ |
|---|---|---|---|---|---|---|---|---|
| PBE [53] | 1.4M | ✗ | 28.35 | 0.766 | 19.96 | 0.139 | 84.63 | 75.72 |
| AnyDoor [6] | 500K | ✗ | 28.03 | 0.767 | 18.85 | 0.137 | 87.74 | 76.91 |
| MimicBrush [7] | 400K | ✗ | 22.50 | 0.784 | 19.41 | 0.111 | 88.21 | 79.11 |
| PBE[⋆] | 1.4M | ✓ | 26.85 | 0.769 | 20.26 | 0.136 | 86.46 | 79.22 |
| AnyDoor[⋆] | 500K | ✓ | 26.62 | 0.772 | 19.13 | 0.135 | 88.73 | 79.37 |
| MimicBrush[⋆] | 400K | ✓ | _19.04_ | _0.791_ | _20.44_ | _0.098_ | **90.87** | **85.65** |
| MimicBrush[†] | 0 | ✓ | 21.12 | 0.785 | 20.39 | 0.106 | 89.48 | 83.26 |
| RoomEditor (Ours) | 0 | ✓ | **18.42** | **0.793** | **21.15** | **0.094** | _90.51_ | _85.47_ |

Table 1: **Quantitative comparison on RoomBench.** PBE, AnyDoor, and MimicBrush are evaluated under three settings: (*i*) released models on large datasets, (*ii*) fine-tuned on RoomBench (marked with ⋆), and (*iii*) re-trained on RoomBench (marked with †). ‡: data volume after filtering via SD [40].

| Method | Fidelity Best (%) | Fidelity Rank↓ | Harmony Best (%) | Harmony Rank↓ | Quality Best (%) | Quality Rank↓ |
|---|---|---|---|---|---|---|
| PBE[⋆] [53] | 5.1 | 3.36 | 18.0 | 2.85 | 4.5 | 3.34 |
| AnyDoor[⋆] [6] | 8.9 | 2.96 | 7.6 | 3.16 | 3.5 | 3.20 |
| MimicBrush[⋆] [7] | 29.4 | 2.11 | 30.5 | 2.16 | 38.3 | 1.80 |
| RoomEditor (Ours) | **56.6** | **1.57** | **43.9** | **1.82** | **53.6** | **1.66** |

Table 2: **User study results.** In each trial, annotators were presented with four images (one from each method) and asked to rank them with respect to fidelity, harmony, and overall quality. "Best (%)" denotes the percentage of cases where a method was ranked first (i.e., perceived as the best under the given criterion), while "Rank↓" indicates the average ranking (lower is better) across all trials.

on our RoomBench, and the re-trained models (denoted with †), whose model parameters are trained by keeping the same configuration as our RoomEditor.

### 5.2.1 Quantitative Comparison

As shown in Table 1, the released models of PBE, AnyDoor, and MimicBrush exhibit limited performance for furniture synthesis despite being trained on large-scale datasets. Notably, the large-scale datasets used for training AnyDoor and MimicBrush primarily consist of video data. The number of training image pairs is estimated to be approximately four times the number of videos, significantly exceeding the scale of our RoomBench dataset. This highlights the necessity of establishing a specific dataset tailored for home design. Fine-tuning on our RoomBench significantly improves the performance of AnyDoor⋆ and MimicBrush⋆ across all metrics. In particular, MimicBrush⋆ achieves the highest CLIP-score and DINO-score.

From a technical standpoint, our RoomEditor, which is trained solely on RoomBench, outperforms competing methods in most quantitative metrics. Specifically, the state-of-the-art MimicBrush[†] is notably inferior to both MimicBrush⋆ and our RoomEditor, demonstrating that our proposed parameter-sharing U-Net architecture is a more effective solution for image composition, especially when large-scale pre-training datasets are not available.

### 5.2.2 User Study

Following the protocol in [7], we conduct a user study to evaluate the perceptual quality of generated images. We let 25 annotators (14 undergraduate students, 7 parents of some of the students, and 4 university faculty or staff members, all informed of the evaluation criteria) rank one hundred randomly selected generation results of different methods based on our benchmark (introduced in section 3) from three aspects: fidelity, harmony, and overall quality. As shown in Table 2, our RoomEditor outperforms competing methods across all aspects. Notably, while MimicBrush⋆ achieves superior results on some objective metrics, it lags behind our approach in real user assessments.

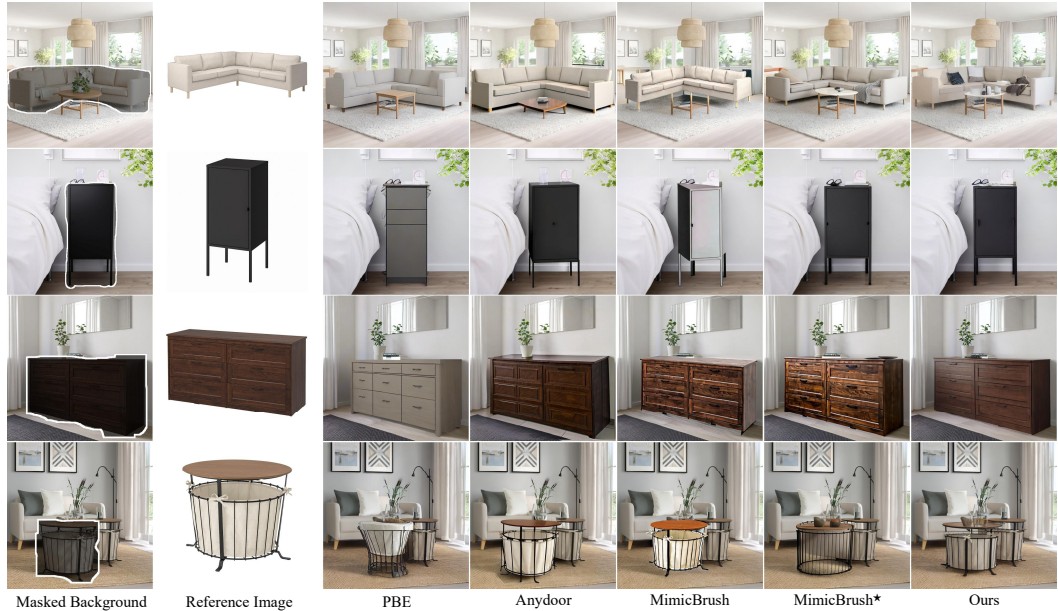

| Masked Background | Reference Image | PBE | Anydoor | MimicBrush | MimicBrush* | Ours |

Figure 4: **Visual results on RoomBench.** PBE and AnyDoor achieve good background harmony but lack object consistency. MimicBrush preserves fidelity but suffers from a copy-paste effect. In contrast, our method ensures both high fidelity and strong background harmony, resulting in realistic and coherent furniture synthesis.

| Method | FID↓ | SSIM↑ | PSNR↑ | LPIPS↓ | CLIP↑ | DINO↑ |
|---|---|---|---|---|---|---|
| PBE [53] | 108.57 | 0.580 | 15.98 | 0.340 | 80.98 | 69.98 |
| AnyDoor [6] | 86.07 | 0.567 | 14.51 | 0.372 | 86.41 | 82.67 |
| MimicBrush [7] | 77.16 | 0.585 | 15.35 | 0.361 | 87.66 | 83.70 |
| RoomEditor (Ours) | **68.78** | **0.594** | **16.43** | **0.304** | **89.62** | **85.04** |

Table 3: **Quantitative evaluation on the DreamBooth dataset.** We compare our method with existing approaches. Despite being trained on only on RoomBench, our method achieves the best results, showcasing strong generalization capability.

### 5.2.3 Qualitative Comparison

As illustrated in Figure 4, PBE [53] fails to keep object identity, leading to significant deviation from the reference. AnyDoor [6] better preserves identity but faces issues of improper scaling, color mismatches, and misplacement. MimicBrush [7] maintains object structure but suffers from artifacts like copy-paste effects and distortion. In contrast, MimicBrush* and our RoomEditor achieve seamless integration, realism, and correct orientation. Given the large-scale dataset used for pre-training MimicBrush*, our RoomEditor is more suitable for furniture synthesis.

### 5.3 Generalization to Diverse Scenes and Objects

We evaluate the generalization performance of our model using the DreamBooth dataset [41], which contains 30 object categories, including backpacks, animals, sunglasses, and cartoon characters. Each sample is annotated with object masks, consistent with the RoomBench dataset. In this experiment, the RoomEditor model is neither re-trained nor fine-tuned on external datasets. We compare it with publicly available models from [53, 6, 7]. As shown in Table 3, our method significantly outperforms existing approaches trained on large-scale datasets. These results demonstrate the strong generalization ability of RoomEditor in synthesizing realistic object placements in unseen environments, indicating its robustness for real-world applications. Qualitative results in Figure 12 further show that RoomEditor naturally integrates diverse objects and scenes.

| Method | All / Train Params (M) | | FID↓ | SSIM↑ | PSNR↑ | LPIPS↓ | CLIP↑ | DINO↑ |
|---|---|---|---|---|---|---|---|---|
| RoomEditor (Ours) | 943 | / 815 | 18.42 | **0.793** | **21.15** | **0.094** | **90.51** | 85.47 |
| RoomEditor+CLIP | 1803 | / 860 | **18.38** | 0.792 | **21.15** | **0.094** | 90.45 | **85.55** |
| Dual U-Net (Frozen $g$)+CLIP | 2432 | / 860 | 21.12 | 0.785 | 20.39 | 0.105 | 89.48 | 83.26 |
| Dual U-Net (Trainable $g$)+CLIP | 2432 | / 1717 | 19.69 | 0.787 | 20.56 | 0.099 | 90.09 | 84.36 |

Table 4: **Ablation Study.** We assess the impact of incorporating CLIP and varying dual U-Net configurations. Adding CLIP improves semantic alignment. Making the reference branch $g$ trainable improves fidelity but increases parameters significantly.

## 5.4 Ablation Study

Since our RoomEditor is highly concise, we conduct ablation study to evaluate its integration with key techniques (e.g., CLIP image encoder and non-shared reference UNet $g$) in existing methods as shown in Table 4. First, we examine the integration of CLIP image encoder, which is commonly adopted to facilitate feature extraction from reference images as in MimicBrush, resulting in RoomEditor+CLIP. Although incorporating CLIP leads to a slight improvement in certain metrics, it introduces a substantial increase in the number of parameters. Next, we analyze the effects of non-shared dual U-Net configurations, where the reference U-Net $g$ is either frozen or trainable. The results show that without our parameter-sharing strategy, both variants exhibit significant performance degradation, particularly when the reference U-Net is frozen. We note that the variant with frozen $g$ is exactly the same as MimicBrush[†]. These findings verify the effectiveness of our parameter-sharing U-Net design for unified feature learning.

## 6 Conclusion

This work advanced furniture synthesis by addressing two key challenges: absence of ready-to-use benchmarks and feature space divergence in dual-branch architectures. Specifically, we collect and release RoomBench, a ready-to-use benchmark for furniture synthesis, while presenting RoomEditor, a parameter-sharing dual U-Net architecture that ensures robust feature alignment. By training our RoomEditor on the collected RoomBench, it achieves state-of-the-art performance for furniture synthesis in terms of geometric coherence and visual fidelity and demonstrates strong generalization ability to diverse scenes. We hope that our work can encourage further research in furniture synthesis.

## Acknowledgements

This work was supported in part by the National Natural Science Foundation of China (62576241 , 62276186, 62172127,U23B2049 and U22B2035).

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

# A    Implementation Details.

**Hyperparameters.**   In our experiments, the inpainting U-Net is initialized with the Stable Diffusion 1.5 [40] inpainting model, which takes a 9-channel input, while the reference U-Net is initialized with the standard Stable Diffusion 1.5 [40] model, which takes a 4-channel input. For the CLIP [38] model, we employ CLIP-H as the image encoder, following [7]. During both training and inference, images are processed to a resolution of $512 \times 512$ by first padding them to a square aspect ratio and then resizing them accordingly. To compute the CLIP-score and DINO-score, we use CLIP ViT-B/32 [38] and DINO ViT-S/16 [3], following the protocols of [41, 53, 6, 7]. For the user study, we recruited 20 annotators to evaluate and rank different methods based on fidelity, harmony, and overall quality.

**Data Augmentation**   To enhance robustness against imperfect masks and domain variations, we apply both image and mask augmentation, as shown in Table 5.

| Type | Augmentation | Description and Parameters |
|------|-------------|---------------------------|
| Image | **Horizontal Flipping** | Applied with a probability of 50%. |
| | **Rotation** | Applied with a probability of 50%, up to 30°. |
| | **Scaling** | Applied with 30% probability, range ±20%. |
| | **Cropping** | Minimum retained ratio: 0.75. |
| Mask | **Perturbation** | 25% probability: Random dilation and erosion for variation. |
| | **Blurring** | 25% probability: Boundary smoothing via blurring and thresholding. |
| | **BBox** | 25% probability: Mask replaced with its bounding box. |
| | **No Augmentation** | 25% probability: Original fine-grained mask retained. |

Table 5: Image and Mask Augmentation Strategies

# B    RoomBench Dataset

In this section, we introduce the construction and characteristics of our RoomBench dataset. As summarized in Table 6, RoomBench covers a broad range of furniture categories with diverse styles and high-quality images, enabling realistic simulation of real-world indoor scenes.

To construct the dataset, we first collected raw images from the internet. The data filtering process, illustrated in Figure 5, was conducted with the assistance of GPT-4o [1], which helped to efficiently filter and organize high-quality image-text pairs. For the training set, we performed precise, pixel-level mask annotation, which supports various mask augmentations—examples of which are shown in Figure 6. In contrast, the testing set was annotated using coarse masks to better reflect real-world conditions, where user-provided annotations are often noisy or imprecise, as depicted in Figure 7.This annotation strategy allows us to evaluate model robustness under imperfect labeling scenarios.

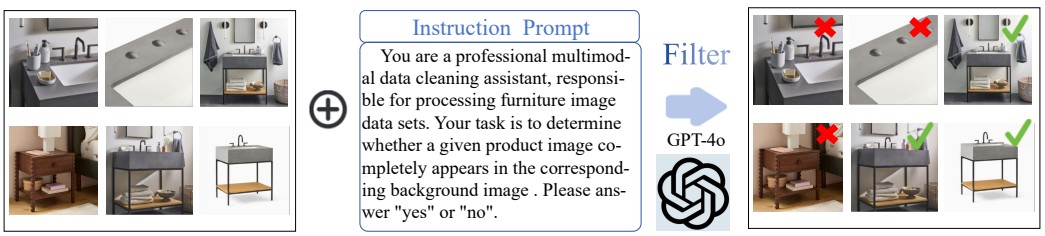

Figure 5: Diagram of data filtering, where images first are labeled as furniture and background images. Then, GPT-4o with specific prompt is used to filter out unmatched and incomplete images.

| Categories | Training Set | Testing Set |
|---|---|---|
| Sofa | 1294 | 61 |
| Lamp | 1280 | 231 |
| Coffee Table | 805 | 24 |
| Wardrobe | 681 | 13 |
| Armchair | 563 | 61 |
| Bed | 460 | 9 |
| Pillow | 440 | 73 |
| Nightstand | 319 | 36 |
| Desk and Chair | 158 | 118 |
| Other | 1298 | 269 |
| Total | 7298 | 895 |

Table 6: Categories and distribution in our RoomBench.

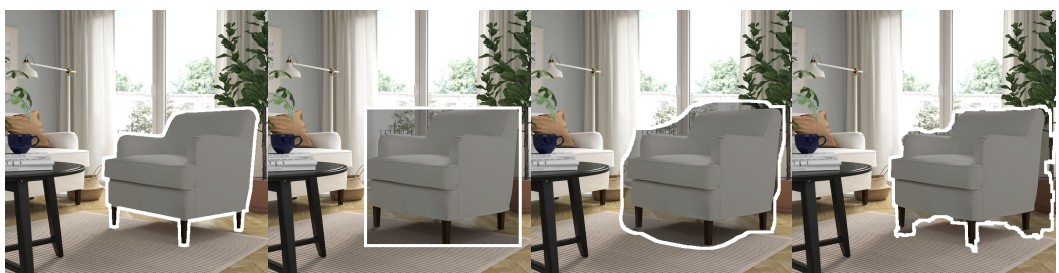

Figure 6: Different forms of data augmentation for precisely annotated mask in training set.

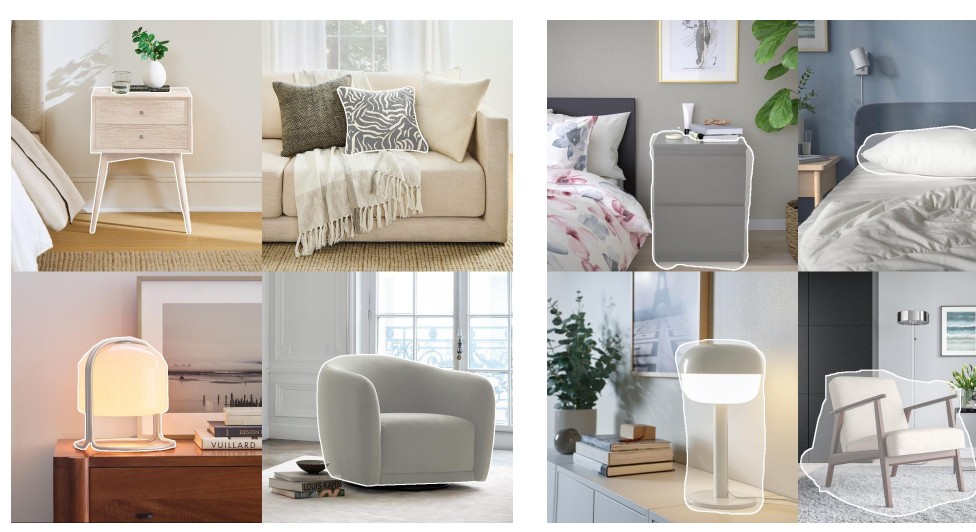

(a) Training samples               (b) Testing samples

Figure 7: In RoomBench, precise masks are annotated for the training samples and can be augmented accordingly. Coarse masks are provided for the testing samples to simulate user interactions.

## C   Additional Experiments

### C.1   Additional Verification Experiments for Eqn. (5) and Eqn. (6)

From subsubsection 4.2.1, we derived Eqn. (5) and Eqn. (6) based on the empirical properties of inpainting models and assumptions about the feature space. To more rigorously justify the validity of these assumptions and properties, we conducted the following experiments.

| Training Loss (Avg.) | Region Merging Loss (Avg.) |
|:---:|:---:|
| $3.9 \times 10^{-2}$ | $4.3 \times 10^{-3}$ |

Table 7: Quantitative comparison between the average training loss and the region merging loss.

| Layer | d0 | d1 | d2 | d3 | d4 | d5 | m0 | u0 | u1 | u2 | u3 | u4 | u5 | u6 | u7 | u8 |
|---|---|---|---|---|---|---|---|---|---|---|---|---|---|---|---|---|
| cos_sim | 0.962 | 0.955 | 0.942 | 0.931 | 0.922 | 0.917 | 0.912 | 0.904 | 0.906 | 0.913 | 0.921 | 0.925 | 0.927 | 0.937 | 0.941 | 0.945 |

Table 8: Cosine similarity across feature layers. Here, `d`, `m`, and `u` correspond to the *downsampling*, *middle*, and *upsampling* blocks of the network, respectively.

For Eqn. (5), the core assumption is that the inpainting diffusion model can preserve the unmasked regions of an image with high fidelity. Therefore, our goal is to demonstrate that the expression

$$\boldsymbol{M}_{\text{bg}} \odot \epsilon_\theta(\boldsymbol{I}_{\text{bg}}^{\text{M}}) + \overline{\boldsymbol{M}_{\text{bg}}} \odot \epsilon_\theta(\mathcal{R}(\boldsymbol{I}_{\text{ref}}^{\text{M}} \mid \boldsymbol{I}_{\text{bg}}^{\text{M}}))$$

can effectively approximate the ground truth (GT) denoising prediction $\epsilon_\theta(\boldsymbol{I}_{\text{gt}})$. To verify this assumption, we conducted experiments using a pretrained diffusion inpainting model on our full test set. During the experiments, we added random noise to the input images and sampled random timesteps $t$. The model was then provided with partial content from the full image (either the background or the object regions). We extracted the predicted noise components from the corresponding regions and computed the $L_2$ loss as the evaluation metric:

$$\left\| \epsilon_\theta(\boldsymbol{I}_{\text{gt}}) - \left( \boldsymbol{M}_{\text{bg}} \odot \epsilon_\theta(\boldsymbol{I}_{\text{bg}}^{\text{M}}) + \overline{\boldsymbol{M}_{\text{bg}}} \odot \epsilon_\theta(\mathcal{R}(\boldsymbol{I}_{\text{ref}}^{\text{M}} \mid \boldsymbol{I}_{\text{bg}}^{\text{M}})) \right) \right\|_2 .$$

The results summarized in Table 7 show that the merged result—obtained by feeding the model with separated regions and combining their corresponding outputs—yields a significantly lower loss than the average training loss (i.e., the average denoising loss during model training, visualized in Figure 3), thereby supporting the validity of the assumption in Eqn. (5).

For Eqn. (6), the underlying assumption is that a property analogous to that in Eqn. (5) also holds at the feature level. To evaluate this hypothesis from a similarity perspective, we adopt *cosine similarity* as the evaluation metric. Specifically, we compute the cosine similarity between the feature representations extracted from different attention layers on the validation set, defined as

$$\text{cos\_sim}\Big( f_l(\boldsymbol{I}_{\text{gt}}), \boldsymbol{M}_{\text{bg}} \odot f_l(\boldsymbol{I}_{\text{bg}}^{\text{M}}) + \overline{\boldsymbol{M}_{\text{bg}}} \odot f_l(\mathcal{R}(\boldsymbol{I}_{\text{ref}}^{\text{M}} \mid \boldsymbol{I}_{\text{bg}}^{\text{M}})) \Big),$$

where $f_l(\cdot)$ denotes the feature map extracted from the $l$-th layer of the model. The results in Table 8 show that cosine similarities remain consistently above 0.9 across most layers, supporting the validity of the assumption in Eqn. (6) and indicating that the feature-level composition effectively preserves the representational coherence of the full image.

## C.2 Generalization to Unseen Datasets: 3D-FUTURE Results

To evaluate the generalization ability of our method on unseen datasets, we conducted cross-dataset experiments using the 3D-FUTURE [14] dataset, which differs from our RoomBench in visual domain and object composition.

We selected 1,020 samples covering 34 furniture categories (randomly 30 samples per category). As the dataset does not provide paired reference images, we generated pseudo-pairs by horizontally flipping reference objects and applying Gaussian-blurred masks to prevent trivial copy-paste solutions.

As shown in Table 9, despite being trained exclusively on RoomBench, our model achieves comparable or superior performance to MimicBrush [7] on most metrics, demonstrating strong generalization to unseen furniture data and novel datasets without fine-tuning. The visualized results are shown Figure 8.

## C.3 Comparison with GPT-4o

To further benchmark our method against a powerful commercial system, we include the closed-source GPT-4o [34] in our evaluation. As a state-of-the-art multimodal model capable of handling multi-

| Method | FID↓ | SSIM↑ | PSNR↑ | LPIPS↓ | CLIP↑ | DINO↑ |
|--------|------|-------|-------|--------|-------|-------|
| MimicBrush [7] | 14.20 | **0.6595** | 20.64 | 0.272 | 80.10 | 64.61 |
| RoomEditor (Ours) | **13.91** | 0.6581 | **20.93** | **0.260** | **80.28** | **66.97** |

Table 9: Cross-dataset evaluation on 3D-FUTURE [14]. Our model outperforms MimicBrush across multiple metrics, demonstrating strong generalization to unseen data distributions.

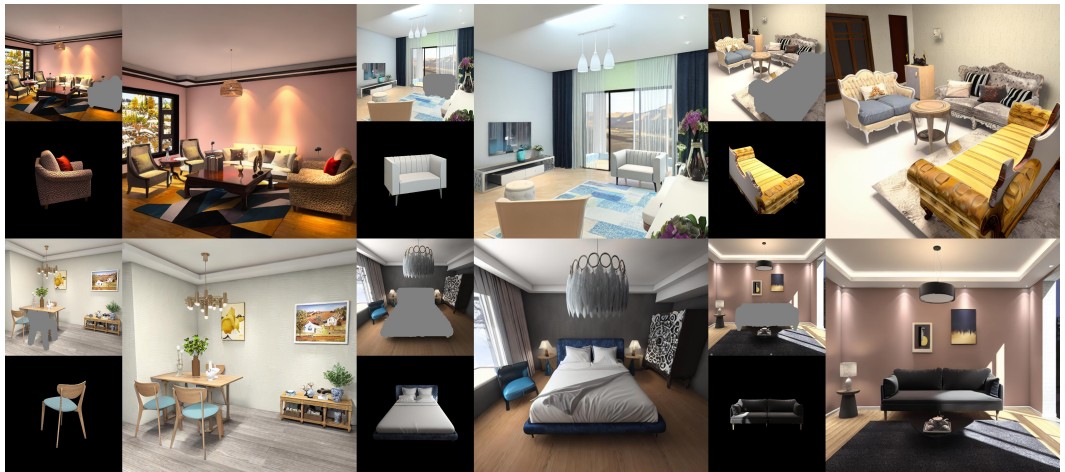

Figure 8: Additional visualization results on the 3D-FUTURE [14] dataset.

| Model | FID (↓) | SSIM (↑) | PSNR (↑) | LPIPS (↓) | CLIP (↑) | DINO (↑) |
|-------|---------|----------|----------|-----------|----------|----------|
| MimicBrush [7] | 56.44 | 0.780 | 19.40 | 0.113 | 88.12 | 79.49 |
| GPT-4o [34] | 72.12 | 0.502 | 13.84 | 0.290 | 87.80 | 83.30 |
| RoomEditor (Ours) | **44.78** | **0.792** | **21.05** | **0.094** | **90.42** | **85.12** |

Table 10: Quantitative comparison with GPT-4o [34] on a 200-sample subset of RoomBench.

image inputs, GPT-4o exhibits strong general-purpose image editing and composition capabilities, making it a valuable reference for assessing high-level visual reasoning and semantic understanding.

For a fair and efficient comparison, we evaluate GPT-4o [34], MimicBrush [7], and our method on a randomly sampled subset of 200 images from the RoomBench dataset. This subset size ensures sufficient scene diversity while keeping closed-source inference costs tractable. A comprehensive set of quantitative metrics is used to capture complementary aspects of visual fidelity and semantic consistency, including FID, SSIM, PSNR, LPIPS, CLIP, and DINO. Among these, CLIP and DINO are particularly informative for assessing object-level fidelity—one of the key challenges in compositional image editing.

The quantitative results are summarized in Table 10. Metrics that emphasize fine-grained spatial and structural consistency, such as FID and LPIPS, show GPT-4o performing notably worse than both MimicBrush and our method. In contrast, its CLIP and DINO scores are comparable to or even higher than those of MimicBrush, reflecting strong global semantic coherence and visually appealing compositions.

While GPT-4o [34] produces visually coherent and semantically meaningful results, it shows two consistent weaknesses. First, **background preservation**: GPT-4o often modifies the surrounding background, introducing shifts in color or texture that reduce spatial consistency and thus worsen FID and LPIPS. Second, **object fidelity**: for certain objects, the generated results exhibit deviations from the reference in geometry or texture, which contributes to lower SSIM and PSNR scores. The visualized results of the comparison are shown in Figure 9.

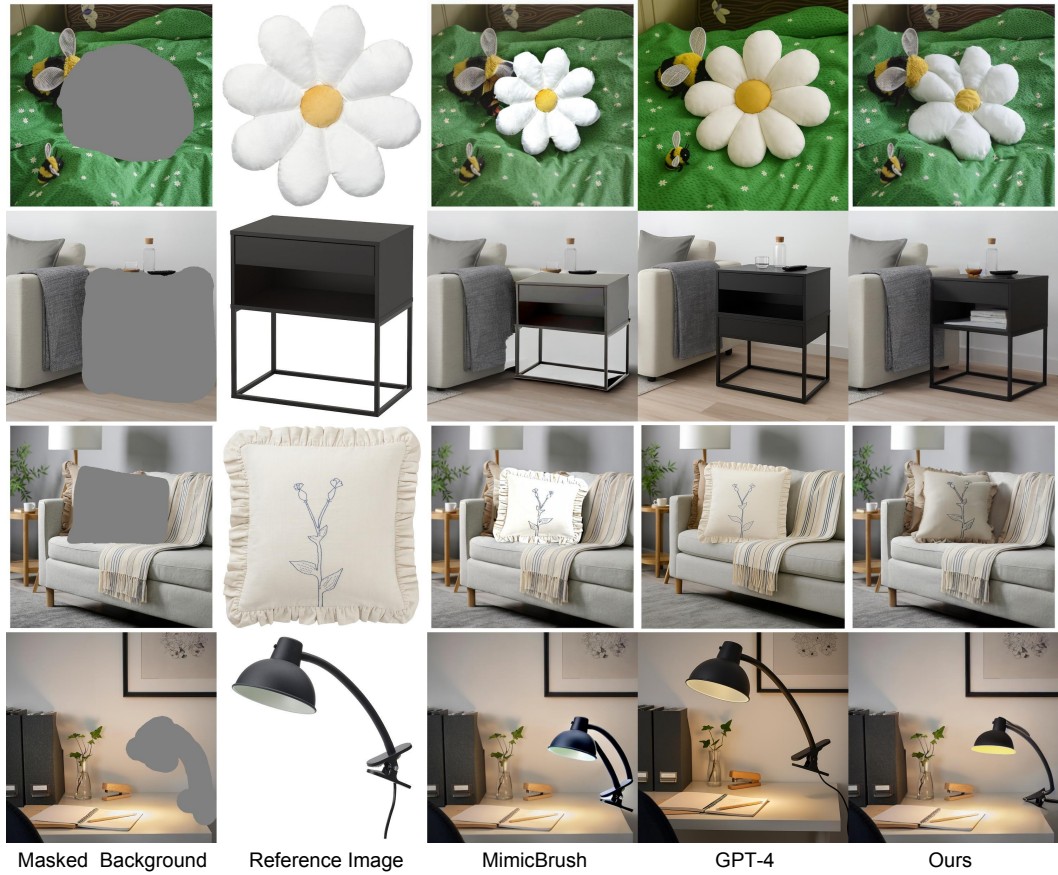

| Masked  Background | Reference Image | MimicBrush | GPT-4 | Ours |

Figure 9: Our RoomEditor compares its visual results with those of Mimicbrush [7] and GPT-4o [34] on RoomBench.

| Method | FID↓ | SSIM↑ | PSNR↑ | LPIPS↓ | CLIP↑ | DINO↑ |
|--------|------|-------|-------|--------|-------|-------|
| MimicBrush [7] | 24.99 | 0.761 | 18.42 | 0.135 | 87.06 | 75.13 |
| RoomEditor (Ours) | **21.16** | **0.766** | **19.83** | **0.120** | **88.90** | **81.46** |

Table 11: Evaluation of robustness to mask quality using bounding boxes as masks. Our method maintains high performance even without accurate shape priors.

### C.4    Robustness to Mask Quality

To further assess the robustness of our model to imperfect masks, we evaluated it using coarse bounding boxes as input masks, thereby removing any explicit shape prior and simulating practical scenarios where precise segmentation is unavailable.

As described in Appendix B, our training strategy already introduces imperfect masks through random perturbations and simplified bounding boxes, encouraging the model to learn shape-consistent reconstruction even under imprecise guidance. During evaluation, we adopt a similar setup by replacing object masks with axis-aligned bounding boxes covering target regions.

As summarized in Table 11, despite lacking detailed shape cues, our method outperforms MimicBrush [7] on most metrics, demonstrating strong robustness to mask variations and high-quality synthesis even with coarse or noisy masks—highlighting its practicality for real-world editing scenarios.

| Light Intensity | 0.6 | 0.8 | 1.0 (base) | 1.2 | 1.4 |
|---|---|---|---|---|---|
| MimicBrush [7] | 70.58 | 71.16 | 71.92 | 70.77 | 69.97 |
| RoomEditor (Ours) | **73.80** | **73.96** | **74.77** | **73.48** | **72.76** |

Table 12: Performance under different lighting intensities. Higher values indicate better performance.

| Model | Params (M) | GPU Mem (GB) | Speed (samples/min) |
|---|---|---|---|
| PBE [53] | 1310 | 3.9 | 14.5 |
| AnyDoor [6] | 2451 | 7.3 | 14 |
| MimicBrush [7] | 2432 | 6.9 | 10.7 |
| RoomEditor (Ours) | **943** | **3.2** | **15.8** |

Table 13: Comparison of computational efficiency across different methods.

## C.5 Robustness to Lighting Changes

To further evaluate the robustness of our method under varying illumination conditions, we conducted additional experiments with systematically adjusted lighting intensities. Since the original test set does not explicitly account for lighting variation, we simulated different illumination conditions by scaling the brightness of background images. Specifically, brightness scaling factors of 0.6 and 0.8 (darker conditions), as well as 1.2 and 1.4 (brighter conditions), were applied to the entire test set.

We compared our method with MimicBrush [7] under these modified lighting conditions, and report the averaged results across all metrics (rescaled to a 0–100 range, with FID and LPIPS inverted for consistency). As shown in Table 12, although both methods slightly degrade under extreme lighting, our approach consistently outperforms [7] across all illumination levels, demonstrating strong robustness and generalizability to lighting variations without explicit modeling.

## C.6 Computational Efficiency Analysis

To provide a fair comparison of computational efficiency across different methods, we report the model size, memory usage, and inference speed on a unified experimental setup. All experiments were conducted on a single NVIDIA A6000 GPU in FP16 precision, using single-image inference with 50 denoising steps.

Table 13 summarizes model efficiency in terms of parameters (Params), GPU memory usage (Mem), and inference throughput (Speed). Our method strikes a favorable balance between performance and efficiency—using significantly fewer parameters and less memory than [7] while achieving about $1.5\times$ higher inference speed—demonstrating its suitability for practical deployment.

## D Additional Visualization Results

In this section, we present additional qualitative results. Figure 10 showcases our results on the RoomBench dataset, while Figure 11 illustrates our performance on the DreamBooth dataset. As observed, our method consistently achieves high fidelity and seamless harmony, demonstrating its effectiveness across both indoor environments and general scenarios. Finally, as shown in Figure 12, qualitative results demonstrate that RoomEditor effectively integrates diverse objects into complex scenes, further validating the utility of RoomBench for real-world applications.

## E Limitations and Future Work

Our method has two primary limitations, which we plan to address in future work.

First, the presence of uncommon furniture items (such as clothes racks and microwaves) results in a long-tail distribution in our collected dataset. Consequently, our model may struggle to capture complex or rare furniture shapes, as the large intra-category variation in geometry makes it difficult for the dataset to fully represent all possible shape patterns. This occasionally leads to distorted

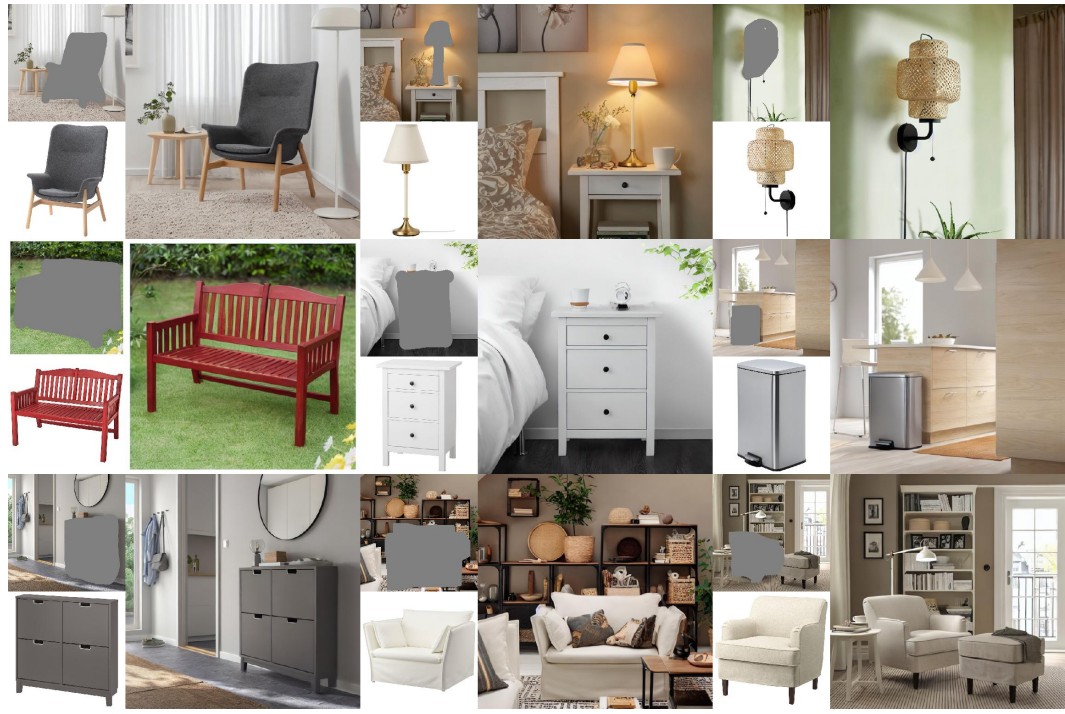

Figure 10: Additional visualization results on the RoomBench dataset, demonstrating the high fidelity and seamless integration of furniture into indoor environments.

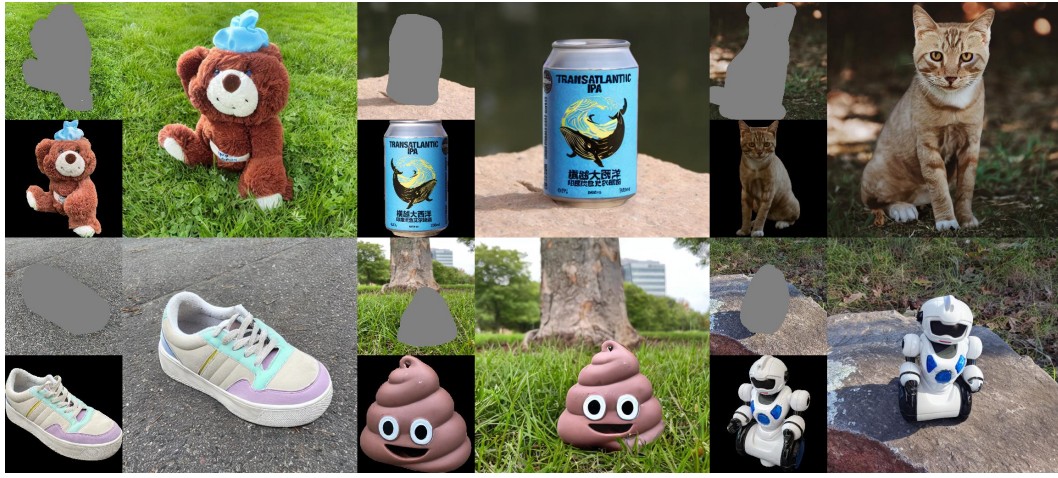

Figure 11: Additional visualization results on the DreamBooth dataset, showcasing the performance of our method in various general scenarios with high fidelity and consistent harmony.

or inconsistent generations, as illustrated in Figure 13. The issue stems from both the insufficient training samples for such items and the challenge of learning robust shape priors under limited data diversity.

Second, our current approach does not explicitly model complex visual phenomena such as object shadows and varying illumination conditions. Since the model edits only the masked region, shadows or lighting effects outside the mask cannot be synthesized. This limitation is inherent to the masked-editing paradigm and is also shared by prior works such as PBE [53], AnyDoor [6], and MimicBrush [7]. These factors may significantly affect visual realism and remain challenging for both our method and existing approaches.

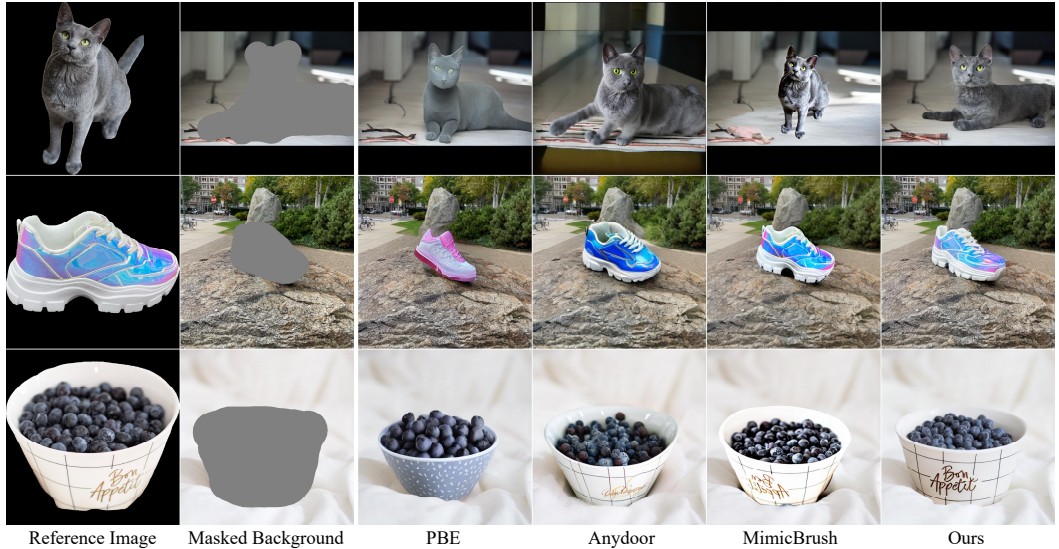

| Reference Image | Masked Background | PBE | Anydoor | MimicBrush | Ours |

Figure 12: Visual comparison on DreamBooth dataset [41].

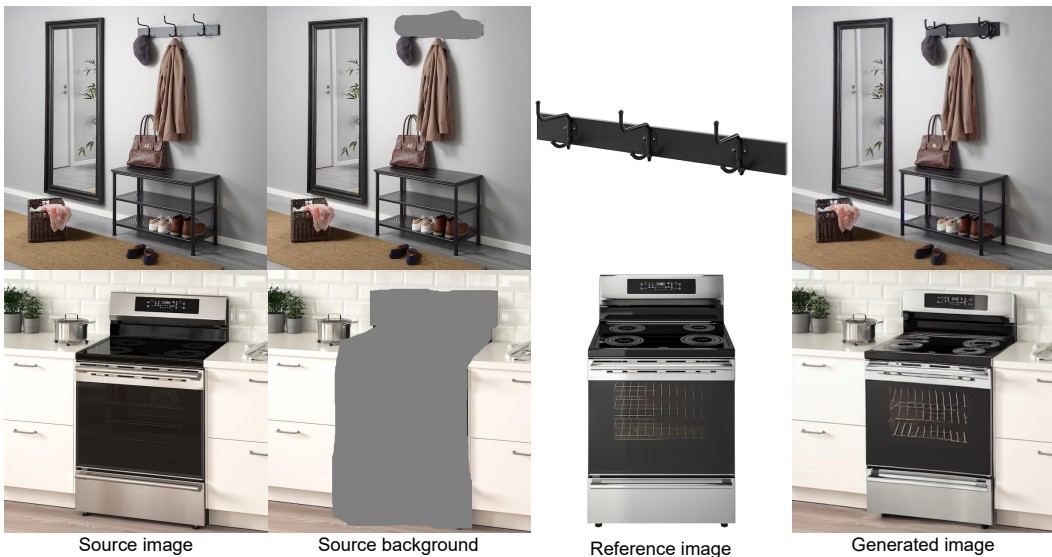

| Source image | Source background | Reference image | Generated image |

Figure 13: Failure cases illustrating distortions and inconsistencies when handling less common furniture items.

Additionally, given the structural differences between our model and recent DiT-based [36] diffusion architectures, adapting our formulation and analytical framework to more advanced DiT diffusion models represents a promising direction for future research.

We leave these aspects for future exploration.

