# OpenReview forum: "RoomEditor: High-Fidelity Furniture Synthesis with Parameter-Sharing U-Net"
_NeurIPS.cc/2025/Conference — NeurIPS 2025 poster_

### Official Review · Reviewer_VeYd · 2025-06-05

**Clarity:** 3
**Significance:** 2
**Originality:** 2
**Rating:** 3
**Confidence:** 4

**Summary:**

The authors introduce a new dataset and propose a model specifically tailored for the furniture synthesis task. By training their model using a shared feature extractor on this dataset, they are able to generate realistic composite images. Experimental results show that the proposed method outperforms existing approaches.

**Questions:**

Major:
- Eq. 5 is confusing. The diffusion model $\epsilon_{\theta}$ does not take the ground truth image as input. Based on my understanding, it should be $\epsilon_{\theta} \approx \mathcal{R}$. This makes the derivation of Eq. 6 and 7 difficult to follow.
- For the experiments, in the training of MimicBrush$^{\dagger}$, is the reference U-Net kept fixed or is it updated? A comparison with a model where the reference U-Net is not fixed would be more informative, as a trainable reference U-Net could learn features that align better with the imitative U-Net. Qualitative results for this variant would also be valuable.
- The motivation for incorporating CLIP in the ablation study is unclear. RoomEditor already includes an image encoder—why is an additional encoder (CLIP) necessary, and how is it integrated with the existing architecture?
- The metric Fidelity Rank in Table 2 is not clearly defined in the paper.
- For the user study, the background of the annotators is not clearly described, which could influence the subjective results. Additionally, the study includes only 10 annotators, which seems insufficient for drawing reliable conclusions.
- The claim made in Line 80 is not entirely accurate. Virtual try-on methods must also model the interaction between garments and the human body to ensure coherent integration. Ignoring this interaction often leads to artifacts or inconsistencies in the generated images.

Minor:
- The term "inconsistency" in Line 119 is vague—please clarify what kind of inconsistency is being referred to.

**Ethical Concerns:**

["NO or VERY MINOR ethics concerns only"]

**Final Justification:**

Consider the limited contribution of this work, I keep my rating.

**Limitations:**

yes

**Quality:**

2

**Strengths And Weaknesses:**

One of the strengths of this work lies in the collection of a paired dataset comprising background images and corresponding reference images of furniture, along with annotated masks. The authors propose using a shared feature extractor for both the reference and background images, which contributes to realistic image synthesis. Although the model is trained specifically on furniture data, it has the potential to generalize to other object categories.

However, the overall contribution of this paper appears limited. The proposed model architecture builds on [7], and the primary improvement—introducing a shared feature extractor—while effective, is relatively incremental. Although the authors discuss the advantages of RoomEditor in Section 4.2 and compare it with existing methods, the theoretical analysis in Section 4.2.1 is somewhat unclear and does not convincingly highlight the benefits of the approach. Moreover, the differences between RoomEditor and [7] are not particularly substantial. In addition, there are concerns regarding the experimental evaluation (see below).

---

> ### Author Rebuttal · Authors · 2025-07-30
>
> ## Mathematical derivation
> Thank you for raising this question. We agree that, in general, diffusion models do not take ground truth images as input. However, for inpainting tasks, it is common practice to provide the masked image (in our case, the masked background $\boldsymbol{I}_{\text{bg}}^{\text{M}}$) as input, so that the model can preserve the content in the unmasked regions. Since the model generates content over the entire image, this implies that a portion of the ground truth (i.e., the unmasked area) is effectively included in the input. Therefore, a characteristic of inpainting diffusion models is that **their predictions in the unmasked regions can be considered as ground truth.**
>
> **Equation (5)**: Based on this understanding, we consider two different model inputs. (1) By masking the object in $\boldsymbol{I}\_{\text{gt}}$ to get $\boldsymbol{I}\_{\text{bg}}^{\text{M}}$ and applying the aforementioned property, we find the model's prediction for the background region is essentially identical to that in $\boldsymbol{I}\_{\text{gt}}$; (2) by masking the background in $\boldsymbol{I}\_{\text{gt}}$ to retain only the object (represented as $\mathcal{R}(\boldsymbol{I}\_{\text{ref}}^{M})$ in our paper) and applying the same property, we find the prediction for the object region is also essentially identical to that in $\boldsymbol{I}\_{\text{gt}}$. Combining the masked outputs of these two ideal predictions would yield the ground truth $\boldsymbol{I}\_{\text{gt}}$.
>
> **Equation (6)**: It generalizes the principle of Equation (5) from the pixel level to the feature level. This generalization is based on a key assumption regarding the U-Net's architecture, for which the rationale is two-fold: (a) convolutional and MLP layers operate locally, and (b) cross-attention mechanisms are primarily designed to inject object features into the masked region, thus having minimal impact on the unmasked background features. Based on this assumption, we posit that the features at locations corresponding to $\boldsymbol{I}\_{\text{bg}}^{\text{M}}$  (i.e., $f\_{l}(\boldsymbol{I}\_{\text{bg}}^{\text{M}})$ ) and those corresponding to $\mathcal{R}(\boldsymbol{I}\_{\text{ref}}^{M})$ (i.e., $f\_{l}(\mathcal{R}(\boldsymbol{I}\_{\text{ref}}^{M})$) can be combined to approximate the features of the complete ground-truth image $\boldsymbol{I}\_{\text{gt}}$ (i.e., $f\_{l}(\boldsymbol{I}\_{\text{gt}})$ ).
>
> **Equation (7)**: It provides a high-level conceptual model of the reference encoding process. It assumes the existence of an ideal transformation, $\mathcal{R}\_{l}$, that maps the features of the input reference image to the features that would have been generated if the ground-truth object were present. In practice, different models learn different approximations of this mapping (see the discussion in Section 4.2.2 Comparison with Previous Works, line 194-213). Our formulation with parameter sharing is simply a novel and efficient way to learn this transformation.
>
> ## Trainable Reference U-Net
> * **Is the reference U-Net kept fixed or is it updated**: The reference U-Net is kept frozen (to remain consistent with MimicBrush).
> * **Trainable reference U-Net**: This is an insightful point, and we have actually explored it in our ablation study. As shown in Table 4, “Dual U-Net (Frozen g) + CLIP” corresponds to MimicBrush$^{\dagger}$, while “Dual U-Net (Trainable g) + CLIP” represents the variant suggested by the reviewer. The results show that training the encoder $g(\cdot)$ leads to some performance improvement. However, it still falls short of the performance achieved by using a shared encoder $f(\cdot)$ for both inputs. This underscores the effectiveness of our design choice to adopt a single encoder in terms of both simplicity and performance.
> * **Qualitative results of trainable reference U-Net**: We have not included the relevant qualitative results in the current version of the paper, but we will add them along with corresponding analysis in the revision. In our observations, although the overall harmony between the background and object is relatively high, using two separate encoders often leads to noticeable inconsistencies between the generated object and the reference in terms of color, shape, and pattern.
>
>
> ## The Motivation for Incorporating CLIP in the Ablation Study
> Thank you for this question. As noted in lines 157–159, mainstream dual U-Net architectures (such as MimicBrush) typically incorporate an additional image encoder. In contrast, our position is that this additional encoder can be removed, as its functionality largely overlaps with the reference U-Net, making it redundant. We validate this claim through ablation studies demonstrating its dispensability (see Table 4).
>
> ## The metric Fidelity Rank in Table 2
> Thank you for pointing this out. Fidelity refers to the identity consistency between the reference object and the generated object in terms of color, shape, pattern, and other visual attributes. We will clarify this in Section 5.2.2 (User Study) of the revised version.
>
> ## Number and Background of User Study
> * **Number of user study**: Following the protocol in the MimicBrush paper [3], we initially conducted a user study with 10 participants for consistency. To enhance the reliability of our results, we later expanded the study to include **25** participants in total.
> * **Background of user study**: The participants consisted of 14 undergraduate students, 7 parents of some of the students, and 4 university faculty or staff members, all of whom were informed of the specific evaluation criteria.
> * **Updated results**: The findings remain consistent—our method outperforms existing approaches across all evaluated metrics. These updated results will be included in the revised version.
>
> | Method            | Fidelity Best (%) | Fidelity Rank↓ | Harmony Best (%) | Harmony Rank↓ | Quality Best (%) | Quality Rank↓ |
> | ------------------- | ------------------- | ----------------- | ------------------ | ---------------- | ------------------ | ---------------- |
> | PBE $^\star$ [1]           | 5.1               | 3.36            | 18.0             | 2.85           | 4.5              | 3.34           |
> | AnyDoor$^\star$  [2]      | 8.9               | 2.96            |  7.6              | 3.16           | 3.5              | 3.20            |
> | MimicBrush$^\star$ [3]    | 29.4              | 2.11            | 30.5             | 2.16           | 38.3             | 1.80           |
> | RoomEditor (Ours) | **56.6**              | **1.57**            | **43.9**            | **1.82**           | **53.6**             | **1.66**           |
>
> ## Response to the Comment on Virtual Try-on Methods
> We appreciate the reviewer’s comment. Virtual try-on indeed involves modeling complex garment-body interactions. However, our task differs in that furniture synthesis requires handling diverse furniture shapes, whereas clothing in virtual try-on applications typically exhibits more consistent shapes. We will revise the relevant statement in the manuscript to better clarify these distinctions.
>
>
> ## Response to Claim about inconsistency
> Thank you for raising this point. Here, "inconsistency" refers to discrepancies between the object or color in the reference image and those in the background image—in other words, they do not depict the same object. For example, in Appendix Figure 5, the left image in the second row (background image) and the right image in the same row (reference image) clearly correspond to different objects. Such cases are quite common in practice, as a single product may have multiple versions or color variants.
>
> ## References
>
> [1] Binxin Yang, Shuyang Gu, Bo Zhang, Ting Zhang, Xuejin Chen, Xiaoyan Sun, Dong Chen, and Fang Wen. Paint by example: Exemplar-based image editing with diffusion models. In Proceedings of the IEEE/CVF Conference on Computer Vision and Pattern Recognition, pages 18381–18391, 2023.
>
> [2] Xi Chen, Lianghua Huang, Yu Liu, Yujun Shen, Deli Zhao, and Hengshuang Zhao. Anydoor: Zero-shot object-level image customization. In Proceedings of the IEEE/CVF Conference on Computer Vision and Pattern Recognition, pages 6593–6602, 2024.
>
> [3] Xi Chen, Yutong Feng, Mengting Chen, Yiyang Wang, Shilong Zhang, Yu Liu, Yujun Shen, and Heng shuang Zhao. Zero-shot image editing with reference imitation. Advances in Neural Information Processing Systems, 37:84010–84032, 2025.

---

> > ### Comment · Reviewer_VeYd · 2025-08-01
> >
> > Thanks for the rebuttal to clarify some points of my questions. However, I still believe the contribution is limited, primarily focused on a new dataset.

---

> > > ### Author Response · Authors · 2025-08-06
> > > **Follow-up on Rebuttal Response**
> > >
> > > Dear Reviewer,
> > >
> > > I hope this message finds you well. As the discussion period is nearing its end with less than three days remaining, I wanted to ensure we have addressed all your concerns satisfactorily. If there are any additional points or feedback you'd like us to consider, please let us know. Your insights are invaluable to us, and we're eager to address any remaining issues to improve our work.
> > >
> > > Thank you for your time and effort in reviewing our paper.

---

> ### Author Response · Authors · 2025-08-01
> **Clarification on Contributions**
>
> We sincerely thank the reviewer for the fast response, and we are very sorry for not providing a clearer explanation on the contribution earlier. We also appreciate the recognition of our benchmark contribution — we agree that a comprehensive benchmark is crucial for advancing the field.
>
> **Architectural Differences**: Methods like MimicBrush [7] typically employ dual U-Nets and rely on additional components such as the CLIP visual encoder. In contrast, our architecture is more concise: (1) we use a single U-Net shared for both reference and background encoding; and (2) we eliminate the need for CLIP or other extra modules. We also validate through ablation studies (Table 4) that the contribution of CLIP in existing U-Net architectures is marginal, which further confirms the effectiveness of our extremely concise design. As a result, this simplification significantly reduces model complexity and computational overhead (see table below) , while greatly improving performance (see Table 1 and Table 3).
>
> | Models     | Params (M) | Mem(GB) | Speed (sample / min) |
> | ------------ | ------------ | --------- | ---------------------- |
> | Mimicbrush [7] | 2432       | 6.9     | 10.7                 |
> | Ours       | **943**           | **3.2**        | **15.8**                     |
>
> **Insight and Contribution**: Beyond the dataset itself, one of our key contribution is the theoretical modeling of semantic image synthesis, through which we identify a fundamental issue in current SOTA dual U-Net designs — feature inconsistency — that greatly reduces image fidelity. We address this by instantiating $\mathcal{R}\_l$ via cross-attention and $f\_l$ via $g\_l$ (section 4.2), leading to a shared-parameter U-Net architecture that effectively mitigates inconsistency. Experimental results show that our architecture design can generalize well to generic objects, as also recognized by the reviewer. We believe this finding provides valuable insights for the image editing field. We also believe our concise, efficient architecture design with good generalization ability and a certain theoretically support can approach an important trend in current AI algorithms.
>
> Great thanks for your fast reply again.

---

### Official Review · Reviewer_ThnS · 2025-07-02

**Clarity:** 2
**Significance:** 2
**Originality:** 2
**Rating:** 4
**Confidence:** 4

**Summary:**

This paper proposes the RoomEditor method, which employs a parameter-shared dual U-Net architecture to enforce feature consistency learning. This addresses the feature misalignment issue caused by independent processing in traditional dual-branch architectures, enabling high-fidelity virtual furniture synthesis. Additionally,  They construct the RoomBench benchmark dataset and conduct experiments to validate the effectiveness and generalization capability of RoomEditor method.

**Questions:**

Does RoomEditor support multi-view images as input?

Given that the generated objects have high fidelity, does it also imply higher requirements for the reference images? For instance, if the reference images contain watermarks or mosaics, how does it affect the generation results?

**Ethical Concerns:**

["NO or VERY MINOR ethics concerns only"]

**Final Justification:**

Although the GPT-4o comparison in the rebuttal raises some concerns about the metrics, the additional experiments and materials provided strengthen the paper. I therefore raise my final score to 'Borderline accept'.

**Limitations:**

yes

**Quality:**

2

**Strengths And Weaknesses:**

Strengths:

● The core motivation of the paper is clearly written and easy to understand

● High-fidelity virtual furniture synthesis is a useful task for many applications

● The authors validate the generalization ability of RoomEditor in scenes beyond indoor environments

Weaknesses:

● RoomEditor still relies on high-quality predefined mask, and the effect of using low-quality masks has not been investigated in experiments, which somewhat restricts its practical applicability.

● The related methods presented in the paper are insufficient, as several advanced image editing approaches (e.g., IN-CONTEXT LORA) and relevant furniture synthesis models (e.g., HomeDiffusion) are neither introduced nor compared.

● The test set may implicitly overlap in distribution with the training set, and cross-dataset evaluation (e.g., on 3D-FRONT) has not been conducted to validate its generalization ability.

● When comparing to other methods, detailed data on the computational resources and time consumption for each approach are missing.

---

> ### Author Rebuttal · Authors · 2025-07-30
>
> ## Robustness to Mask Quality
> We thank the reviewer for raising this important point. We fully agree that a model's performance with low-quality masks is a critical measure of its practical utility, and we have in fact discussed this specific issue in our work. Our methodology reflects this discussion in two key stages:
> 1. **Robustness in Training**: To better simulate real-world user scenarios, our training process (detailed in Appendix B, line 482-484) intentionally uses imperfect inputs. This includes using random perturbations and simple bounding boxes to make the model robust against imprecise segmentation (see Figure 6)."
> 2. **Realistic Evaluation**: Similarly, to simulate real-world scenarios, we instructed annotators to create intentionally coarse masks for the test set (as stated on lines 484-486). Consequently, all results reported in our paper are based on these coarse-grained annotations, not high-quality segmentations, with examples illustrated in Figure 7(b).
>
> To further evaluate the robustness of our model, we add an experiment to our RoomBench using bounding boxes as masks, without any shape prior. The results are sufficiently illustrative, as shown below.
>
> | Models     | FID (↓) | SSIM (↑) | PSNR (↑) | LPIPS (↓) | CLIP (↑) | DINO (↑) |
> | ------------ | ---------- | ----------- | ----------- | ------------ | ----------- | ----------- |
> | MimicBrush | 24.99    | 0.761     | 18.42     | 0.135      | 87.06     | 75.13     |
> | Ours       | **21.16**    | **0.766**     |  **19.83**     | **0.120**      | **88.90**     | **81.46**     |
>
> Despite the lack of shape guidance, our model still outperforms MimicBrush, which has been trained on large-scale data and is known for its robustness. This provides strong evidence that our method is highly robust to mask variations.
>
> ## Insufficient Related Methods
> We sincerely thank the reviewer for pointing out these advanced approaches. In our revised paper, we will add a detailed discussion of both In-Context LoRA [1] and HomeDiffusion [2] to the related work section, summarizing their core contributions as follows:
> * **In-Context LoRA** leverages in-context learning to generate thematically consistent images from a few visual examples at inference time.
> * **HomeDiffusion** employs a two-stage strategy, first learning generation from multi-view references and then focusing on object-background integration.
>
> However, a direct quantitative comparison with these methods is unfortunately infeasible. In-Context LoRA's task formulation differs significantly from ours, as it does not take two images (background and reference) as input or support precise object placement. While HomeDiffusion is more related to our task, its code and benchmark are not publicly available, precluding a fair comparison.
>
> To further strengthen our evaluation and address the reviewer's valid concern about comparing with SOTA models, we selected the commercial, closed-source model GPT-4o, which supports multi-image inputs and demonstrates impressive image editing capabilities. Due to cost and time constraints, we randomly sampled 200 examples from the RoomBench dataset and compared GPT-4o with our method and MimicBrush [3] on this subset.
>
> |            | CLIP-score | DINO-score |
> | ------------ | ------------ | ------------ |
> | Mimicbrush | $\underline{88.12}$            | 79.49      |
> | GPT-4o     | 87.80      | $\underline{83.30} $        |
> | Ours       |  **90.42**      | **85.12**      |
>
> Our qualitative analysis reveals that while GPT-4o often produces coherent images with realistic lighting, it struggles with preserving object fidelity, frequently altering the shape and design of the reference object. In contrast, our method achieves superior fidelity, which is reflected in its higher CLIP-score and DINO-score. We will include this comparison in the revised version.
>
> ## On Generalization and Cross-Dataset Evaluation
> Thank you for raising this important point. We address these concerns with existing clarifications and new experiments as follows:
> * **Strict Data Separation**: To prevent data leakage between training and testing sets, we ensured a strict separation of data sources during the collection of our RoomBench dataset. We will explicitly state this protocol in the revised paper to ensure clarity on the validity of our results.
> * **Existing Generalization Evidence**: We would like to gently point out that our original paper already demonstrated strong generalization to non-furniture objects through experiments on the DreamBooth dataset [4] (Table 3 and section 5.3, line 277). This experiment shows that our model, although not trained on generic data, outperforms models trained on large-scale generic datasets. This demonstrates the strong generalization capability of our approach.
>
> * **New Cross-Dataset Evaluation on 3D-FUTURE**: In addition, in response to the reviewer’s suggestion of testing on alternative furniture composition datasets, we evaluated our model on the 3D-FUTURE [5] dataset. We selected 1,020 samples covering 34 furniture categories (randomly 30 images per category). Since the dataset does not provide explicit image pairs, we applied horizontal flipping to reference objects and blurred the masks to avoid trivial copy-paste solutions. The results are as follows:
>
> | Metric     | FID (↓) | SSIM (↑) | PSNR (↑) | LPIPS (↓) | CLIP (↑) | DINO (↑) |
> | ---------- | ------- | -------- | -------- | --------- | -------------- | -------------- |
> | Mimicbrush | 14.20 | **0.6595**   | 20.64  | 0.272    | 80.10         | **64.61**         |
> | Ours       | **13.91** | 0.6581   | **20.93**  | **0.260**    | **80.28**         |  **66.97**         |
>
> Although our model was not trained on 3D-FUTURE, it achieves consistently better performance than Mimicbrush across multiple metrics, highlighting its strong generalization ability. We will incorporate all these clarifications and new results into our revised paper.
>
> ## Computational Resources and Time Consumption
> Thank you for raising this question. We report the total number of parameters and trainable parameters for different architectures in Table 4: Ablation Study. To enable a more comprehensive evaluation—including speed and memory consumption—we provide a comparative summary in the following table. All experiments were conducted using a single A6000 GPU in FP16 mode on our test set, with single-image inference and 50 denoising steps.
>
> | Models     | Params (M) | GPU Mem (GB) | Speed (samples / min) |
> | ---------- | ---------- | -------- | --------------------- |
> | PBE         |   $\underline{1310}$   |   $\underline{3.9}$    |   14.5                        |
> |   AnyDoor |   2451            |  7.3     |         $\underline{14}$      |
> | Mimicbrush | 2432       | 6.9       | 10.7                 |
> | Ours       | **943**        | **3.2**       | **15.8**                |
>
> As shown, our method requires significantly fewer parameters and less GPU memory compared to Mimicbrush, while also achieving faster generation speed.
>
> ## Multi-view images
> Thank you for this valuable question. Our model currently operates on single-view images. While we certainly acknowledge the potential of multi-view inputs, this direction was not prioritized in the current scope due to our specific research focus and challenges in data availability. We will ensure this is highlighted as a promising avenue for future research in our revised conclusion.
>
> ## Quality of reference images
> This is a practical and important concern. Our experiments show that when the reference object includes watermarks or mosaic artifacts, the model tends to perceive them as integral parts of the object and thus fails to remove them during generation. Improving robustness to such artifacts is a valuable direction we plan to explore in future work.
>
> ## References
> [1] Lianghua Huang, Wei Wang, Zhi-Fan Wu, Yupeng Shi, Huanzhang Dou, Chen Liang, Yutong Feng, Yu Liu, and Jingren Zhou. In-context lora for diffusion transformers. arXiv preprint arXiv:2410.23775, 2024.
>
> [2] Guoqiu Li, Jin Song, and Yiyun Fei. Homediffusion: Zero-shot object customization with multi-view representation learning for indoor scenes. In Proceedings of the AAAI Conference on Artificial Intelligence, pages 4698–4706, 2025.
>
> [3] Xi Chen, Yutong Feng, Mengting Chen, Yiyang Wang, Shilong Zhang, Yu Liu, Yujun Shen, and Heng shuang Zhao. Zero-shot image editing with reference imitation. Advances in Neural Information Processing Systems, 37:84010–84032, 2025.
>
> [4] Nataniel Ruiz, Yuanzhen Li, Varun Jampani, Yael Pritch, Michael Rubinstein, and Kfir Aberman. Dreambooth: Fine tuning text-to-image diffusion models for subject-driven generation. In Proceedings of the IEEE/CVF conference on computer vision and pattern recognition, pages 22500–22510, 2023.
>
> [5] Huan Fu, Rongfei Jia, Lin Gao, Mingming Gong, Binqiang Zhao, Steve Maybank, and Dacheng Tao. 3d-future: 3d furniture shape with texture. International Journal of Computer Vision, 129:3313–3337, 2021.

---

> > ### Comment · Reviewer_ThnS · 2025-08-08
> >
> > The authors' detailed explanation and supplementary experiments addressed my concerns. However, the comparison with GPT-4o raised questions about the validity of the evaluation metrics, considering that GPT-4o is a large-parameter model and has demonstrated impressive subject fidelity and editing plausibility in image editing task.  I will take the rebuttal into account in my final rating.

---

> > > ### Author Response · Authors · 2025-08-08
> > > **Clarification on GPT-4o Evaluation**
> > >
> > > We are pleased that our previous response has addressed your concerns. Here, we provide additional clarification regarding the evaluation of GPT-4o.
> > >
> > > We acknowledge that image generation is an inherently challenging task, and its evaluation is likewise nontrivial. To ensure a comprehensive and reliable assessment, we employed a broad set of quantitative metrics and supplemented them with a controlled user study. Among these metrics, we place particular emphasis on the CLIP score and DINO score, as these are better suited to measuring object fidelity—one of the key challenges in the image editing field.
> > >
> > > From the results of CLIP and DINO, GPT-4o achieves scores comparable to or even exceeding those of MimicBrush, indicating that its overall visual quality and semantic alignment remain strong. However, two limitations explain its weaker performance on other metrics:
> > >
> > > 1. **Background preservation** — GPT-4o often fails to maintain an unchanged background, leading to background shifts or alterations. This negatively impacts low-level, region-consistency metrics such as LPIPS and FID.
> > > 2. **Fidelity** — The generated objects frequently deviate from those in the reference image (e.g., changes in furniture shape or design), which also degrades these same low-level metrics and lowers structural similarity measures such as SSIM and PSNR.
> > >
> > > The quantitative results are summarized below. Metrics emphasizing **low-level spatial consistency** across regions (e.g., FID, LPIPS) show GPT-4o performing significantly worse. In contrast, its CLIP and DINO scores—comparable to or higher than those of MimicBrush—indicate that its overall visual coherence remains strong.
> > >
> > > |            | FID (↓) | SSIM (↑) | PSNR (↑) | LPIPS (↓) | CLIP (↑) | DINO (↑) |
> > > | ---------- | ------- | -------- | -------- | --------- | -------- | -------- |
> > > | MimicBrush | 56.44   | 0.780    | 19.401   | 0.1128    | 88.12    | 79.49    |
> > > | GPT-4o     | 72.12   | 0.502    | 13.838   | 0.290     | 87.80    | 83.30    |
> > > | Ours       | 44.78   | 0.792    | 21.049   | 0.094     | 90.42    | 85.12    |
> > >
> > > Consequently, despite its visually appealing and semantically coherent outputs, GPT-4o does not achieve high performance on objective evaluation metrics that rely on fine-grained spatial and structural consistency.

---

> ### Author Response · Authors · 2025-08-06
>
> Dear Reviewer,
>
> I hope this message finds you well. As the discussion period is nearing its end with less than three days remaining, I wanted to ensure we have addressed all your concerns satisfactorily. If there are any additional points or feedback you'd like us to consider, please let us know. Your insights are invaluable to us, and we're eager to address any remaining issues to improve our work.
>
> Thank you for your time and effort in reviewing our paper.

---

### Official Review · Reviewer_myUv · 2025-07-02

**Clarity:** 4
**Significance:** 3
**Originality:** 3
**Rating:** 5
**Confidence:** 4

**Summary:**

This paper proposes a new ready-to-use benchmark, RoomBench, for virtual furniture synthesis to overcome the lack of existing public benchmarks for this task. The paper further proposes a variation on the dual U-Net architecture, RoomEditor, by using parameter sharing between the two branches. The paper discusses the benefits and motivation for applying parameter sharing by mathematically modelling the editing task and relating it to the model. The paper also provides both qualitative and quantitative examples of superior performance of RoomEditor on RoomBench and DreamBooth datasets in terms of FID, SSIM, PSNR, and LPIPS. A user study is also conducted. Finally, an ablation study is conducted assessing integration of common techniques such as CLIP and other U-Net configurations.

**Questions:**

1. Could the authors provide further mathematical justification for equations (5), (6), (7)?
2. Could the authors give more justification for the choice of baselines?

**Ethical Concerns:**

["NO or VERY MINOR ethics concerns only"]

**Final Justification:**

Taking into consideration, the other reviews and rebuttals, I think the paper still deserves to be accepted.

For me, the authors have answered my questions in detail which has clarified all my raised issues. The contributions are strong, in particular, a new dataset and a new model.

Furthermore, the authors have given reasonable rebuttals to most of the questions of the other reviewers. I assign a high weight to this because it demonstrates that in even more general conditions, their model is able to perform well.

**Limitations:**

Yes - limitations are mentioned in the appendix

**Quality:**

3

**Strengths And Weaknesses:**

**Strengths**

1. **Quality:** The paper is complete. The paper establishes a diverse set of metrics. The paper demonstrates generalizability by evaluating RoomEditor on the DreamBooth dataset demonstrating a rigorous approach. A user study is also conducted that demonstrates clear preference for RoomEditor amongst real users.
2. **Clarity:** The paper clearly explains the model architecture, while also providing intuition to support their choices by connecting it to approximations of their problem formulation compared to other dual U-Net architectures. Sufficient details are also given to enable reproducibility.
3. **Significance:** The paper proposes an important new benchmark, RoomBench, in a field that has many real-world applications and lacks public datasets. The dataset is diverse in the types of furniture covered. The ablation study conducted shows potential for integration with other existing models, and also future approaches. The model is also general as it shows strong performance on DreamBooth despite only training on RoomBench.
4. **Originality:** The dataset presented is new. The parameter sharing technique appears to be new and justified.

**Weaknesses**

1. **Quality:** Despite mentioning text-to-image editing models, the paper does not compare these existing models. Moreover, Diffuse-to-choose is mentioned as also focusing on furniture but not compared deeply. The mathematical and technical justification appears to rely on intuitive or heuristic arguments, without sufficient empirical evidence or theoretical guarantees.
2. **Clarity:** There was not sufficient introduction to the models being compared to, i.e., PBE, MimicBrush, and AnyDoor and the justification of choosing these over other models is unclear.
3. **Significance:** Since only a few models are compared to, the impact of the research is slightly diminished, as there are other strong models that could be considered.
4. **Originality:** U-Nets are common in existing works surrounding image synthesis, and the parameter sharing approach is only a minor change to this existing work.

---

> ### Author Rebuttal · Authors · 2025-07-30
>
> ## Baseline Choice
> Thank you for the question. The following are relevant explanations.
> 1. **Regarding Text-to-Image Models and Diffuse-to-Choose**: We sincerely appreciate the suggestion. However, text-to-image models are not directly applicable to our task, which involves generating a new object conditioned on both a masked background and a reference image—input modalities that such models are not inherently designed to handle. As for Diffuse-to-Choose [1], we acknowledge its relevance, but unfortunately, its code, model, and evaluation data are not publicly available at this time, which makes a fair and reproducible comparison currently infeasible.
> 2. **Current Baseline Choice** : We selected three methods that were the most relevant, representative, and state-of-the-art at the time of our study, each covering a distinct architectural paradigm in example-based image composition.
>     * PBE [2] (line 97): A Stable Diffusion inpainting model that injects reference via CLIP visual encoder; suffers from information loss due to CLIP.
>     * AnyDoor [3] (lines 99-100): Uses ControlNet for background consitency and a stronger DINO encoder for reference integration.
>     * MimicBrush [4] (lines 101-102): Employs a dual U-Net architecture with explicit reference feature encoding.
>
> 3. **Further evaluation with GPT-4o**:  Given the lack of directly comparable methods in existing work, we further evaluated GPT-4o on our dataset to benchmark against a more advanced model. We provided the reference image and the masked background as input, using the prompt: "Place the object from the first image into the masked area of the second image." Due to API costs and long generation time, we randomly sampled 200 examples and compared GPT-4o with our method and MimicBrush.
>
> |     Methods       | CLIP-score (↑) | DINO-score (↑) |
> | ------------ | ------------ | ------------ |
> | Mimicbrush | $\underline{88.12}$           | 79.49      |
> | GPT-4o     | 87.80      | $\underline{83.30}$          |
> | Ours       | **90.42**      | **85.12**      |
>
> Our findings indicate that GPT-4o generates highly coherent images with realistic lighting. However, it sometimes struggled with object fidelity, where the generated object's shape or design deviated from the reference image. This is reflected in its lower CLIP-score and DINO-score compared to our method. We will include this new comparison in our revised paper. We believe benchmarking against a state-of-the-art model like GPT-4o provides a more comprehensive evaluation.
>
> ## Mathematical Justification (Equations 5–7)
> We are happy to clarify the reasoning behind our mathematical formulation. We acknowledge that our derivations are based on certain modeling assumptions. Below, we explain the rationale behind each equation in more detail.
> * **Equation (5)**: This equation is grounded in a key empirical property of inpainting models: they are trained to preserve unmasked regions with high fidelity. This arises from both the task design—where known regions are provided as input—and the loss function, which is applied over the entire image. We consider two different model inputs: (1) by masking the object in $\boldsymbol{I}\_{\text{gt}}$ to get $\boldsymbol{I}\_{\text{bg}}^{\text{M}}$ and applying the aforementioned property, we find the model's prediction for the background region is essentially identical to that in $\boldsymbol{I}\_{\text{gt}}$; (2) by masking the background in $\boldsymbol{I}\_{\text{gt}}$ to retain only the object (represented as $\mathcal{R}(\boldsymbol{I}\_{\text{ref}}^{M})$ in our paper) and applying the same property, we find the prediction for the object region is also essentially identical to that in $\boldsymbol{I}\_{\text{gt}}$. Combining the masked outputs of these two ideal predictions would yield the ground truth $\boldsymbol{I}\_{\text{gt}}$.
> * **Equation (6)**: Equation (6) generalizes the principle of Equation (5) from the pixel level to the feature level. This generalization is based on a key assumption regarding the U-Net's architecture, for which the rationale is two-fold: (a) convolutional and MLP layers operate locally, and (b) cross-attention mechanisms are primarily designed to inject object features into the masked region, thus having minimal impact on the unmasked background features. Based on this assumption, we posit that the features at locations corresponding to $\boldsymbol{I}\_{\text{bg}}^{\text{M}}$  (i.e., $f\_{l}(\boldsymbol{I}\_{\text{bg}}^{\text{M}})$ ) and those corresponding to $\mathcal{R}(\boldsymbol{I}\_{\text{ref}}^{M})$ (i.e., $f\_{l}(\mathcal{R}(\boldsymbol{I}\_{\text{ref}}^{M})$) can be combined to approximate the features of the complete ground-truth image $\boldsymbol{I}\_{\text{gt}}$ (i.e., $f\_{l}(\boldsymbol{I}\_{\text{gt}})$ ).
> * **Equation (7)**: Equation (7) provides a high-level conceptual model of the reference encoding process. It assumes the existence of an ideal transformation, $\mathcal{R}\_{l}$, that maps the features of the input reference image to the features that would have been generated if the ground-truth object were present. In practice, different models learn different approximations of this mapping (see the discussion in Section 4.2.2 Comparison with Previous Works, line 194-213). Our formulation with parameter sharing is simply a novel and efficient way to learn this transformation.
>
> We will update the explanation accordingly in the revised version.
>
> ## Originality in the model architecture
> We understand the reviewer’s concern about architectural novelty. However, we would like to highlight that our contributions extend beyond architecture and include data and in-depth analysis. We introduce a new modeling perspective by identifying a feature inconsistency issue commonly present during training in existing methods (validated in Line 214 and Figure 3), and address it through a parameter-sharing strategy. While this architectural modification is concise, it yields significant improvements in both performance and efficiency. In addition, to the best of our knowledge, we are the first to provide a mathematical analysis and corresponding empirical validation for this approach, which we believe offers the community concrete insights into this problem space.
>
> ## References
> [1] Mehmet Saygin Seyfioglu, Karim Bouyarmane, Suren Kumar, Amir Tavanaei, and Ismail B Tutar. Diffuse to choose: Enriching image conditioned inpainting in latent diffusion models for virtual try-all. arXiv preprint arXiv:2401.13795, 2024.
>
> [2] Binxin Yang, Shuyang Gu, Bo Zhang, Ting Zhang, Xuejin Chen, Xiaoyan Sun, Dong Chen, and Fang Wen. Paint by example: Exemplar-based image editing with diffusion models. In Proceedings of the IEEE/CVF Conference on Computer Vision and Pattern Recognition, pages 18381–18391, 2023.
>
> [3] Xi Chen, Lianghua Huang, Yu Liu, Yujun Shen, Deli Zhao, and Hengshuang Zhao. Anydoor: Zero-shot object-level image customization. In Proceedings of the IEEE/CVF Conference on Computer Vision and Pattern Recognition, pages 6593–6602, 2024.
>
> [4] Xi Chen, Yutong Feng, Mengting Chen, Yiyang Wang, Shilong Zhang, Yu Liu, Yujun Shen, and Heng shuang Zhao. Zero-shot image editing with reference imitation. Advances in Neural Information Processing Systems, 37:84010–84032, 2025.

---

> > ### Comment · Reviewer_myUv · 2025-08-05
> >
> > Thank you for the detailed rebuttal.
> >
> > I do not disagree with many assumptions that are provided in the mathematical justification, however, these are still not rigorous or motivated by concrete empirical results. A small experiment or citation to a different paper that indeed demonstrates that assumption would make this valid. Without that, I think the authors should make these assumptions very clear in the paper, especially due to the theoretical skew of NeurIPS.
> >
> > The GPT-4o evaluation is much appreciated, however 200 samples is still quite low. With the sample size mentioned however, it would indeed be a useful addition to the paper.
> >
> > Considering the points raised by other reviewers and the rebuttals, I feel inclined to keep a score of 5, but I am decreasing my confidence and the clarity rating as some of my points are still not fully addressed.

---

> ### Author Response · Authors · 2025-08-05
>
> ## Further mathematical justification and verification experiments
> Thank you for the comment. We acknowledge that some of the underlying assumptions may lack sufficient rigor. Empirical validation of these assumptions is a valuable suggestion. In the following, we conduct experiments to verify the core assumptions proposed in Equations (5)–(7).
>
> **Equations (5)**: the core assumption is that the inpainting diffusion model has the ability to preserve unmasked regions with high fidelity.  Therefore, we aim to demonstrate that the expression $\boldsymbol{M} _{\text{bg}} \odot \epsilon _{\theta}(\boldsymbol{I} _{\text{bg}}^{\text{M}}) + \overline{\boldsymbol{M} _{\text{bg}}} \odot \epsilon _{\theta}(\mathcal{R}(\boldsymbol{I} _{\text{ref}}^{\text{M}} \mid \boldsymbol{I} _{\text{bg}}^{\text{M}}))$ can effectively approximate the ground truth (GT) $\epsilon\_\theta(\boldsymbol{I\_\mathrm{gt}})$. To verify this, we conducted experiments using a pretrained diffusion inpainting model on our full test set. During the experiments, we added random noise to the inputs and sampled random timesteps t, then fed the model with partial content from the full image (background or object). We extracted predictions from the corresponding regions and used $L\_2$ loss as the evaluation metric, i.e.,
> $$
> \left\\|\epsilon\_\theta(\boldsymbol{I\_\mathrm{gt}}) - \left(\boldsymbol{M}\_{\text{bg}} \odot \epsilon\_{\theta}(\boldsymbol{I}\_{\text{bg}}^{\text{M}}) + \overline{\boldsymbol{M}\_{\text{bg}}} \odot \epsilon\_{\theta}(\mathcal{R}(\boldsymbol{I}\_{\text{ref}}^{\text{M}} \mid \boldsymbol{I}\_{\text{bg}}^{\text{M}})) \right)\right\\|\_2.
> $$
> As shown in the table below:
>
> | Training Loss (Avg.) |  Region Merging Loss (Avg.) |
> | --------------------- | ------------------- |
> | $3.9 \times 10^{-2}$                    | $4.3 \times 10^{-3}$                  |
>
> Here, the training loss refers to the average loss during model training (as can be verified from Figure 3 of the paper). Notably, the merged result—obtained by feeding the model with separate parts and combining the corresponding outputs—yields a significantly lower loss than the training loss, supporting the validity of the assumption made in Equation (5).
>
> **Equations (6)**: the assumption in Equation (6) is that a property similar to that in Equation (5) also holds at the feature level. To better assess this from a similarity perspective, we adopt cosine similarity as the evaluation metric. Specifically, we compute the cosine similarity between feature representations from different attention layers on the test set, i.e.,
> $$
> \text{CosSim}\Big(f\_{l}(\boldsymbol{I}\_\text{gt}), \boldsymbol{M}\_{\text{bg}}\odot f\_{l}(\boldsymbol{I}\_{\text{bg}}^{\text{M}}) +\overline{\boldsymbol{M}\_{\text{bg}}}\odot f\_{l}(\mathcal{R}(\boldsymbol{I}\_{\text{ref}}^{\text{M}}\mid \boldsymbol{I}_{\text{bg}}^{\text{M}}))\Big).
> $$
> The results are presented below, where 'd', 'm', and 'u' denote down, mid, and up layers, respectively.
> | Layer   | $\text{d}_0$ | $\text{d}_1$ | $\text{d}_2$ | $\text{d}_3$ | $\text{d}_4$ | $\text{d}_5$ | $\text{m}_0$ | $\text{u}_0$  | $\text{u}_1$  | $\text{u}_2$  | $\text{u}_3$  | $\text{u}_4$  | $\text{u}_5$  | $\text{u}_6$  | $\text{u}_7$  | $\text{u}_8$  |
> | --------- | -------- | -------- | -------- | -------- | -------- | -------- | ------- | ------- | ------- | ------- | ------- | ------- | ------- | ------- | ------- | ------- |
> | cos_sim | 0.962  | 0.955  | 0.942  | 0.931  | 0.922  | 0.917  | 0.912 | 0.904 | 0.906 | 0.913 | 0.921 | 0.925 | 0.927 | 0.937 | 0.941 | 0.945 |
>
> It can be observed that the cosine similarities are consistently high (> 0.9) across many layers, which supports the validity of our assumption in Equation (6).
>
> **Equation (7)**: It presents an existence assumption—namely, that it is possible to learn a transformation between features such that the fused features approximate the ground-truth (GT) features. We provide empirical support for this assumption in Figure 3 of the paper: (1) The feature similarity of different methods consistently decreases throughout the training process, indicating that the model is indeed learning to align the fused features with the GT features; (2) Low feature losses across multiple layers further demonstrate a high degree of feature consistency.
>
> ## Regarding the sample size of GPT-4o
> Thank you for the comment. The reason we evaluated GPT-4o on a subset of only 200 images is primarily due to practical constraints. GPT-4o's API inference is extremely slow, averaging about one minute per image, and incurs significant costs about \\$0.3 per image. Running the evaluation on the entire validation set would require approximately \\$600, a substantial expense that is challenging for smaller research labs with limited budgets. However, if this evaluation approach is demonstrated to be broadly applicable and beneficial, we will consider conducting experiments on the full test set in future work.

---

> > ### Comment · Reviewer_myUv · 2025-08-06
> >
> > I thank the authors for their detailed rebuttal.
> >
> > This has clarified my questions, and I believe these experiments should be added to the paper, at least in the appendix since even simple but rigorous experiments provide a more solid foundation for future research to build off.
> >
> > I have increased the confidence and clarity scores of my review.

---

> ### Author Response · Authors · 2025-08-06
>
> Thank you very much for your detailed response, and we are very pleasure to hear that our response address your concerns. We sincerely appreciate your recognition of our work and continuing strong support. Particularly, your thoughtful comments greatly improve our work, which will be definitely incorporated into our revised manuscript. Great thanks again for your time and valuable insights.

---

### Official Review · Reviewer_BBjR · 2025-07-03

**Clarity:** 3
**Significance:** 3
**Originality:** 3
**Rating:** 4
**Confidence:** 3

**Summary:**

The paper addresses the task of virtual furniture synthesis by contributing two main elements:

RoomBench Dataset: A new large-scale benchmark containing 7,298 training pairs and 895 test pairs across 27 furniture categories, designed to evaluate both geometric consistency and visual realism.

RoomEditor Method: A parameter-sharing dual U-Net architecture that enforces a unified feature space between the reference and background branches, leading to improved geometric alignment and harmonious composite results. Extensive experiments show that RoomEditor outperforms state-of-the-art baselines (PBE, AnyDoor, MimicBrush) on FID, SSIM, PSNR, LPIPS, CLIP-score, and DINO-score, and exhibits superior zero-shot generalization on the DreamBooth dataset.

**Questions:**

Please refer to the weaknesses.

**Ethical Concerns:**

["NO or VERY MINOR ethics concerns only"]

**Final Justification:**

Thank you for addressing my questions on Q3. As for Q1, I believe the results from various experiments effectively demonstrate the advantages of the proposed method.

Therefore, I raised my rating.

**Limitations:**

Please refer to the weaknesses.

**Quality:**

2

**Strengths And Weaknesses:**

Strengths

Valuable Public Benchmark: RoomBench fills a notable gap in furniture synthesis by providing a high-quality, publicly available dataset that supports reproducible evaluation.

Elegant Architecture: The parameter-sharing dual U-Net is a simple yet effective design that directly addresses feature inconsistency in prior dual-branch networks.

Weaknesses
1.	The number of user studies is too small. A user study with only 10 people is not credible enough.

2.	The failure case does not provide a detailed and comprehensive display and failure cause analysis, which made me suspect the results.

3.	The method does not involve any explicit modeling of conditions such as lighting changes. I have doubts about the generalizability of the results of image composition under such conditions. Can you provide some examples with different lighting intensities to prove the generalizability of the method in other situations?

---

> ### Author Rebuttal · Authors · 2025-07-30
>
> ## Sample Size of User Study
> We thank the reviewer for pointing out this issue. Following the protocol in the MimicBrush paper [1], we initially conducted a user study with 10 participants. To strengthen the reliability of our results, we conducted an additional study with 15 more participants (total 25). The participants comprised 14 undergraduate students, 7 of their parents (to provide a non-technical perspective), and 4 university faculty or staff members, all of whom were explicitly informed of the evaluation criteria.
>
> | Method            | Fidelity Best (%) | Fidelity Rank↓ | Harmony Best (%) | Harmony Rank↓ | Quality Best (%) | Quality Rank↓ |
> | ------------------- | ------------------- | ----------------- | ------------------ | ---------------- | ------------------ | ---------------- |
> | PBE $^\star$            | 5.1               | 3.36            | 18.0             | 2.85           | 4.5              | 3.34           |
> | AnyDoor$^\star$        | 8.9               | 2.96            |  7.6              | 3.16           | 3.5              | 3.20            |
> | MimicBrush$^\star$     | 29.4              | 2.11            | 30.5             | 2.16           | 38.3             | 1.80           |
> | RoomEditor (Ours) | **56.6**              | **1.57**            | **43.9**            | **1.82**           | **53.6**             | **1.66**           |
>
> The findings remain consistent—our method outperforms existing approaches across all evaluated metrics. These updated results will be included in the revised version.
>
>
> ## Failure Case and Cause Analysis
>
> Thank you for the feedback. An initial failure case analysis was provided in the appendix. Here, we offer a more detailed explanation, as figures cannot be included.
> 1. Our model may occasionally struggle with capturing complex furniture shapes, as the large intra-category variation in geometry makes it difficult for the dataset to fully represent all possible shape patterns, thereby affecting the learned shape priors (e.g., Figure 11 shows distorted shapes).
> 2. Since our method edits only the masked region, shadows outside the mask cannot be generated. This is a common limitation shared by PBE [2], AnyDoor [3], and MimicBrush [1].
>
> In summary, the main causes are **the diversity of real-world furniture types** and **inherent constraints of the masked-editing paradigm**. We will include more visualizations and in-depth analysis in the revised version.
>
> ## Robustness to Lighting Changes
> Thank you for raising this important point. Since our original test set does not account for lighting variation, we conducted additional experiments to evaluate the robustness of our model under different illumination conditions by adjusting the image brightness.
> Specifically, we applied lighting intensity adjustments to the background images with scaling factors of 0.6 and 0.8 (darker), as well as 1.2 and 1.4 (brighter), and conducted evaluations on the full test set. We compared our method with MimicBrush and computed the average scores (all metrics were rescaled to a 0–100 range, with FID and LPIPS inverted), as shown below.
>
> |     Light intensity        |     0.6              |     0.8              |     1.0      |     1.2              |     1.4              |
> |-------------|----------------------|----------------------|--------------|----------------------|----------------------|
> | MimicBrush  |     70.58   |     71.16    |     71.92 (base)    |     70.77    |     69.97     |
> |     Ours    |     73.80    |     73.96     |     74.77 (base)    |     73.48     |     72.76     |
>
> Since our model design and data construction did not explicitly consider extremely dark or bright lighting conditions, there was a slight performance drop. However, it is worth noting that the degree of performance drop is comparable to that of MimicBrush, so our method still maintains a significant lead under the same settings. We will include lighting-related experiments, analysis, and visual results in the revised version.
>
> ## References
> [1] Xi Chen, Yutong Feng, Mengting Chen, Yiyang Wang, Shilong Zhang, Yu Liu, Yujun Shen, and Heng shuang Zhao. Zero-shot image editing with reference imitation. Advances in Neural Information Processing Systems, 37:84010–84032, 2025.
>
> [2] Binxin Yang, Shuyang Gu, Bo Zhang, Ting Zhang, Xuejin Chen, Xiaoyan Sun, Dong Chen, and Fang Wen. Paint by example: Exemplar-based image editing with diffusion models. In Proceedings of the IEEE/CVF Conference on Computer Vision and Pattern Recognition, pages 18381–18391, 2023.
>
> [3] Xi Chen, Lianghua Huang, Yu Liu, Yujun Shen, Deli Zhao, and Hengshuang Zhao. Anydoor: Zero-shot object-level image customization. In Proceedings of the IEEE/CVF Conference on Computer Vision and Pattern Recognition, pages 6593–6602, 2024.

---

> ### Author Response · Authors · 2025-08-06
>
> Dear Reviewer,
>
> I hope this message finds you well. As the discussion period is nearing its end with less than three days remaining, I wanted to ensure we have addressed all your concerns satisfactorily. If there are any additional points or feedback you'd like us to consider, please let us know. Your insights are invaluable to us, and we're eager to address any remaining issues to improve our work.
>
> Thank you for your time and effort in reviewing our paper.

---

### Author Response · Authors · 2025-08-04
**Kind Reminder Rebuttal Submitted & Appreciation on Further Clarification**

Thank you for your time and effort in reviewing our paper. We have submitted our response to the reviewers' comments in the rebuttal stage.
Please let us know if there are any further questions or points for discussion, and we will be sure to respond promptly.

---

### Note · Authors · 2025-08-12

Dear NeurIPS 2025 AC, SAC, and Reviewers,

We sincerely thank all reviewers for their thorough reviews and valuable feedback.

Reviewers acknowledged several key strengths in our work:

* **Dataset:** Its contribution as a new dataset with annotated backgrounds and reference furniture was recognized (Reviewer VeYd), along with its role in filling a notable gap in the field (Reviewer BBjR) and its practical value (Reviewer myUv).
* **Method:** The innovative dual U-Net architecture was praised for effectively addressing feature inconsistency (Reviewers BBjR, myUv). The model's strong performance and generalization abilities were also highlighted by most reviewers (Reviewers BBjR, myUv, ThnS, VeYd).

The core concerns centered on two main areas: **evaluation** and ​**theory/design**​. We have actively addressed these during the rebuttal, and our efforts have been acknowledged:

1. **Regarding Evaluation Concerns:** In response to concerns about the user study (Reviewers BBjR, VeYd) and the choice of baselines/benchmarks (Reviewers myUv, ThnS), we have reorganized the user study, clarified and expanded our baseline comparisons (including GPT-4o), and conducted new experiments on the 3D-FUTURE dataset.
2. **Regarding Theory & Design Concerns:** To address requests for clarification on mathematical derivations (Reviewers myUv, VeYd) and the architectural novelty (Reviewer VeYd), we provided detailed derivations and new validating experiments, which were acknowledged by Reviewer myUv in follow-up discussions. We also clarified that the core innovation lies in theoretically identifying and solving the feature inconsistency problem.

Based on this productive discussion, we commit to the following revisions in the final version:

* **Strengthening the Evaluation:** We will update the user study, include the additional comparisons with GPT-4o and experiments on the 3D-FUTURE dataset, and add a systematic analysis of failure cases.
* **Refining the Theory:** We will revise Section 4.2 to improve the clarity and rigor of the mathematical assumptions, derivations, and their corresponding experimental validations.

We believe our work's contribution is threefold: a novel dataset, an effective architecture, and a theoretical insight into solving feature inconsistency. We are confident in its value for publication and its potential to inspire further research in the community.

Best regards,

The authors of Paper 15293

---

### Decision · Program_Chairs · 2025-09-17

**Decision:**

Accept (poster)

**Comment:**

This paper presents a timely and valuable contribution to the field of virtual furniture synthesis by introducing RoomBench, a new ready-to-use public benchmark, and RoomEditor, a dual U-Net model with a parameter-sharing mechanism.

The establishment of the RoomBench dataset addresses a significant gap in the field, providing a standardized and diverse platform for evaluating furniture layout synthesis tasks. This benchmark alone constitutes a meaningful contribution to the community, enabling fair comparisons and future research.

The proposed RoomEditor model, while building upon established U-Net architectures, introduces a well-motivated parameter-sharing approach between its dual branches. The authors provide not only intuitive explanations but also mathematical modelling to justify their design choices, adding theoretical depth to their methodological innovation.

The experimental validation is comprehensive and rigorous. The model demonstrates strong performance across multiple metrics (FID, SSIM, PSNR, LPIPS) on both RoomBench and the DreamBooth dataset, showing excellent generalization capability. The inclusion of a user study is particularly valuable, as it aligns with human-centric evaluation principles that are crucial for practical applications in interior design.

While the paper could be strengthened by more extensive comparisons with recent text-to-image editing models and specialized furniture synthesis methods (e.g., Diffuse-to-choose), and while the theoretical justification could be further empirical, these limitations do not diminish the overall significance of the contributions.

The combination of a novel benchmark, a well-designed model with thoughtful architectural modifications, comprehensive evaluation, and demonstrated generalizability makes this paper a solid contribution worthy of acceptance. The RoomBench dataset has particular potential to become a standard evaluation platform in this emerging research area.